

# Radiative impact of increased middle atmospheric water vapour in the aftermath of the Hunga 2022 volcanic eruption at two locations in the Northern Hemisphere

Alistair Bell[a,b], Axel Murk[a,b], and Gunter Stober[a,b]

[a]Institute of Applied Physics, University of Bern, Bern, Switzerland
[b]Oeschger Centre for Climate Change Research, University of Bern, Bern, Switzerland

**Correspondence:** Alistair Bell (alistair.bell@unibe.ch)

**Abstract.** Increases in middle atmosphere water vapour as a result of the 2022 Hunga volcanic eruption have now been detected almost globally, with above-average mixing ratios predicted to persist until around 2032. Changes in the middle atmosphere water vapour volume mixing ratio impact chemical reactions in this section of the atmosphere, can result in more favorable conditions for polar stratospheric and mesospheric cloud formation, and have a significant radiative effect on the middle atmosphere and below. For this reason, precise radiative transfer calculations are important to make accurate and precise assessments of changes to both long-wave and short-wave fluxes, and how this may impact the heating rates at different heights in the atmosphere. In this study, water vapour profiles from two microwave radiometers deployed at two different latitudes in Europe are used to analyse changes in water vapour in the aftermath of the Hunga volcano, and a line-by-line radiative transfer model is used to analyse the thermal impact of this increase over Bern, Switzerland, and Ny-Ålesund, Svalbard.

## 1 Introduction

The 2022 Hunga (also commonly referred to as Hunga-Tonga or Hunga Tonga-Hunga Ha'apai [HTHH]) volcanic eruption, having a volcanic explosivity index of at least VEI-5 (Terry et al., 2022) would have been a notable global event considering simply the eruptive power, with the eruption having comparable energy to the Krakatoa volcanic eruption Matoza et al. (2022). The event caused various phenomena in the atmosphere; launching atmospheric waves traveling around the Earth up through all layers including the thermosphere (Wright et al., 2022; Stober et al., 2023; Themens et al., 2022; Vadas et al., 2023b, a; Stober et al., 2024). What gave the event even more importance on a global scale, however, was the amount of water that was vapourised and then injected into the middle atmosphere, in a plume that reached above $50\,\mathrm{km}$ asl during the peak of the eruption (Millan et al., 2022) (GOES satellite images revealed that the ash plume reached 53 km (Carr et al., 2022)). It has been estimated that the mass of water vapour injected into the stratosphere during the eruption was $146\,\mathrm{Tg}$- equal to approximately $10\,\%$ of total global stratospheric water vapour, or around the same amount as enters the stratosphere from all sources globally during one year.

The water vapour in the middle atmosphere is relatively stable because of the few sinks in this part of the atmosphere. When a rare event such as the injection of water vapour into the middle atmosphere occurs, such as the case with the Hunga eruption,





this means that perturbations can have effects lasting for several years. Jucker et al. (2023) have found from climate model simulations that the increased water vapour from the Hunga volcano is likely to last eight years or more. The climatic impact of this has thus sparked interest in the scientific community.

Aside from the injection of water vapour into the stratosphere, a moderate amount (totalling 0.6-0.7 Tg) of sulphur dioxide was also emitted during the entire eruption sequence (Carn et al., 2022). Whilst increased water vapour has a warming net radiative effect at the surface due to increased long-wave absorption, sulphur dioxide, which forms sulfate aerosols in the atmosphere, has the effect of reducing the amount of short-wave radiation that reaches the surface and, thus, results in a negative radiative impact at the surface.

Whilst Jenkins et al. (2023) have suggested that the volcano would have a net warming effect on the global surface, meaning that global temperature anomalies could exceed the threshold $1.5\,^{\circ}\mathrm{C}$ limit, other authors have suggested that the dominant effect of the eruption was due to the aerosol, which would result in a surface cooling effect. Sellitto et al. (2022) estimated the radiative effect in the weeks after the eruption to be negative at the surface $[-1.7\,\mathrm{W\,m^{-2}}]$. Schoeberl et al. (2023b) estimated that the mean surface temperature change from the combination of a reduction in short-wave flux and an increase in long-wave flux would be negative, with the effects causing cooling of $0.038\,^{\circ}\mathrm{C}$ for clear skies and $0.021\,^{\circ}\mathrm{C}$ for all skies (i.e. including cloud cover) in the southern hemisphere.

Whilst providing reasonable estimates of radiative flux calculations on a global scale, and being able to investigate numerous changes due to atmospheric composition changes, one drawback of studies using general circulation models (GCMs) is that the radiation schemes used are heavily parameterised to keep computational resources reasonable. GCMs such as the Whole Atmosphere Community Climate Model (WACCM) utilise correlated-k distribution methods, grouping wavelengths with similar absorption characteristics to allow a radiative transfer scheme to represent the entire short-wave or long-wave spectrum without having to calculate absorption coefficients across every absorption line. It has been postulated that such radiative transfer schemes are accurate to within $0.20\,\mathrm{W\,m^{-2}}$ to $0.23\,\mathrm{W\,m^{-2}}$ of more accurate line-by-line calculations, and within $0.15\,\mathrm{K\,d^{-1}}$ for stratospheric heating rates (Iacono et al., 2008).

The effect of a changing radiative balance affects not only the temperature at the surface. With increased long-wave emission or short-wave absorption, atmospheric layers may cool down or heat up in response. Around the altitude and latitude of the maximum water vapour anomaly in the weeks following the eruption ( $38\,\mathrm{hPa}$ to $10\,\mathrm{hPa}$; 30°S to 5°N ), it was found that there were correlations with negative temperature anomalies (Basha et al., 2023). Another point that appears to be of crucial importance for the radiative balance, is how the mixing ratio of ozone responds to changes in water vapour. (Basha et al., 2023) also found that increases in water vapour corresponded to decreases in ozone, with anomalies of up to $-0.5\,\mathrm{ppmv}$ being collocated with increases in water vapour of around $2\,\mathrm{ppmv}$. This is explained by the production of hydroxyl radicals, which have a net effect of reducing ozone through reactions involving hydrogen oxides. In work unrelated to the HTHH volcano, it has also been found that cooling induced by increases in the amount of water vapour in the lowermost stratosphere has a significant effect on the subtropical and eddy-driven jet streams which can affect weather patterns in the troposphere (Charlesworth et al., 2023).





Several months after the eruption, a portion of the water vapour plume had been transported across the equator into the
northern hemisphere, which has been proposed to have been partly due to equatorial Rossby waves generated by long-wave
cooling of stratospheric air by the increased water vapour itself (Schoeberl et al., 2023a). This thermal anomaly was perhaps
increased due to the separation of the water vapour, which ascended in altitude in this time period, and the aerosol layer, which
descended Schoeberl et al. (2022). Already in April 2022, observations had been made of the increase in water vapour anomaly
at Manu Loa at $19.5°$ N (Nedoluha et al., 2023). As of November 2023, almost all the water vapour initially injected into the
middle atmosphere remained there, albeit at higher altitudes on average (Nedoluha et al., 2024), which resulted in the highest
recorded mesospheric water vapour since operations began in the 1990's at Manu Loa.

Continuous observations have also been made by two radiometers designed and operated by the University of Bern. These
two radiometers in Bern, Switzerland, and Ny-Ålesund, Svalbard, have continuously measured water vapour profiles in the
middle atmosphere from 2006 and 2015 respectively, until today. This provides a valuable resource for estimating the increases
in water vapour caused by the HTHH volcano in the three years following eruption at two locations. Two methods based on the
same line-by-line radiative transfer simulator are then run to make precise calculations of the long-wave flux anomalies caused
by the increase in water vapour. One of these methods is also used to calculate local heating rates from both short-wave and
long-wave impacts of the water vapour changes in Bern, Switzerland.

## 2 Instruments

### 2.1 Ground-Based Radiometric Observations

Several microwave radiometers have been developed, deployed, and maintained by the University of Bern to measure at-
mospheric composition and temperature, particularly for the middle atmosphere (Bell et al., 2025; Fernandez et al., 2015;
Sauvageat et al., 2022; Krochin et al., 2022; Shi et al., 2023). These long-term observations are ideal for investigating the im-
pact of sudden disruptive events such as the HTHH eruption. In this study, we leverage data from two water vapour radiometers
currently operated next to Bern at the mid-latitudes and at Ny-Ålesund on Svalbard covering also the high-latitudes (Schranz
et al., 2020).

### 2.1.1 MIAWARA

The MIddle Atmosphere WAter vapour RAdiometer (MIAWARA) is a 22 GHz radiometer designed for long-term monitoring
of water vapour in the middle atmosphere. It has been operated by the Institute of Applied Physics (IAP) at the University of
Bern since April 2002 and was installed at the Zimmerwald Observatory near Bern, Switzerland, in 2006.

The observational frequency is centred on the 22.235 GHz water vapour absorption line. A balancing calibration scheme
using a reference view that optimizes noise and linearity (Forkman et al., 2003). Hot and cold targets are used, with the sky
at a high elevation angle serving as the cold target. Currently, two spectrometers are used. One FFT spectrometer: the Aquiris





AC-240 with 61 kHz channel spacing and a Software-defined radio (SDR) USRP X310 with two input receivers, and 12.2 kHz channel spacing, and a 14-bit ADC for reduced bias and improved linearity.

The retrieval of water vapour profiles is handled by the ARTS (atmospheric radiative transfer software) Buehler et al. (2018), with an optimal estimation algorithm, utilising the Levenburg-Marquart algorithm for iterative updates, and a priori profiles taken from ECMWF analysis climatology and MLS water vapour climatology. The time series from 2011 has been recently reprocessed, from which this analysis has been conducted (Bell et al., 2025).

### 2.1.2   MIAWARA-C

The MIddle Atmospheric WAter vapour RAdiometer for Campaigns (MIAWARA-C) was based on the original design of the MIAWARA, but specifically designed for measurement campaigns (Tschanz et al., 2013) (data access at: Bell (2025)). Developed by the University of Bern, MIAWARA-C is more compact and can be operated remotely. The observational frequency is also centred on the 22.235 GHz absorption line, and makes use of the balancing calibration routine. In addition, the radiometer was upgraded to include a dual-polarization receiver. This allows the instrument to utilise independent measurements from two
receiver channels corresponding to the same air mass, which are used together to reduce the measurement noise. After several campaigns, including Bern (47°N, 7°E) and Sodankylä (67°N, 27°E) from 2010 to 2013, the radiometer was moved to the AWIPEV research base in Ny-Ålesund, Svalbard (79°N, 12°E) where it has been based since September 2015 (Schranz et al., 2018, 2019; Shi et al., 2024).

    The retrieval process is similar to that explained in Bell et al. (2025), but has one key difference in that a priori water vapour
profiles for the lower stratosphere contain information from the ECMWF forecasts corresponding to the retrieval time.

### 2.2   Space-Borne

Space-borne instruments can be an immensely useful resource for tracking and monitoring water vapour in the middle atmosphere: the key advantage that is offered when compared to ground-based instrumentation is the horizontal coverage of measurements. Depending on orbit geometry and instrument field of view the entire globe can be observed. However, the
revisit time for specific geographic locations is much lower when contrasted to ground-based sensors.

### 2.2.1   Aura-MLS

The Microwave Limb Sounder (MLS) aboard the Aura satellite is a microwave radiometer with five independent receivers that provide high-resolution spectral data. It retrieves at least 15 atmospheric constituents (including water vapour) and temperature profiles at heights from $5\,\text{km}$ to $80\,\text{km}$ (Waters et al., 2006; Schwartz et al., 2008). MLS is often used as a reference for other
middle atmosphere measurements due to its frequent and co-located measurements covering all latitudes from 82° S to 82° N. It has also been an invaluable resource for tracing the water vapour plume evolution in the aftermath of the Hunga eruption (attached video material documenting the global spread from MLS).



The MLS science team has implemented a "duty cycle" for the 190-GHz observations, which include $H_2O$, $N_2O$, HCN, and upper stratospheric $HNO_3$, to conserve the remaining life of the instrument. This involves reactivating the subsystem for
approximately six days each month, with possible adjustments based on the aging of the receiver (NASA JPL, 2024). The instrument is also not expected to make further observations beyond Spring 2026 at the latest due to its de-orbiting. However, technical issues can even shorten the remaining lifetime substantially.

### 2.2.2 ACE-FTS

The ACE-FTS (Atmospheric Chemistry Experiment Fourier Transform Spectrometer) aboard the SCISAT-1 satellite uses high-
resolution solar occultation to infer atmospheric constituents, including water vapour, and measures vertical profiles from the cloud tops up to 150 km. The instrument performs two soundings per orbit, during sunrise and sunset, which means that the temporal coverage provided by the satellite is much reduced compared to Aura-MLS, and revisit times are variable. For water vapour, the retrieval height is limited to $101\,\mathrm{km}$. The quality of the v2.2 $H_2O$ dataset assessed by Sheese et al. (2017) indicated a consistent positive bias in the measurements from the lower stratosphere up into the mesosphere, typically around 3% to 8%
compared to other datasets.

### 3 Water Vapour Evolution

Much has been written about the evolution of the initial water vapour plume (Schoeberl et al., 2023a; Nedoluha et al., 2024). To summarise this: shortly after the eruption, the water vapour plume was initially confined mostly to the Southern Hemisphere. Within a month, there was a significant northward movement of the plume across the equator, which was attributed to infrared
cooling effects associated with the high levels of water vapour, which likely induced equatorial Rossby waves. These waves helped to push the water vapour across the equatorial barrier into the Northern Hemisphere. During the descending phase of the Quasi Biennial Oscillation (QBO), there was also a notable cross-equatorial transport of water vapour, which occurred along-side changes in the Brewer-Dobson circulation. The shift in the QBO and associated circulation dynamics further transported water vapour across the equator, extending its impact and distribution. Almost all of the water vapour that was initially injected
into the stratosphere remained above the tropopause two years after the eruption.

With MLS data, the transport of the water vapour anomaly is visible to as far north as $80°$. This should affect the water vapour profiles above the observation sites in Bern, Switzerland and Ny-Ålesund, Svalbard. An animated graphic has been produced showing the meridional, and vertical transport of water vapour around the globe (see Supplementary Material).

A comparison of the observations from MIAWARA and MIAWARA-C to MLS and ACE-FTS was performed to verify
the retrievals, shown in figure 1; whilst MIAWARA over-estimates and the ACE-FTS underestimates relative to Aura-MLS at 1hPa, the pattern of observations remains constant and all show significant increases in water vapour anomalies from mid-2022. Water vapour measurements above Ny-Ålesund exhibit for the MIAWARA-C and ACE-FTS slightly lower values compared to MLS retrievals, but again resemble the overall seasonal morphology with reasonable agreement. This aligns with other cross-comparisons with these instruments and reanalysis data performed before (Shi et al., 2023). It can be noted that for this



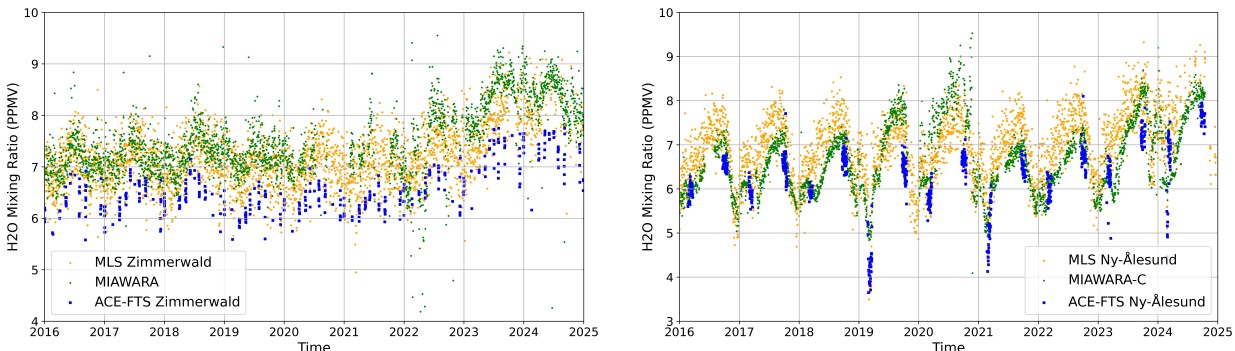

**Figure 1.** Water vapour at 1hPa retrieved from the MIAWRARA, MLS and ACE-FTS instruments between 2018 and March 2024.

comparison, all ACE-FTS measurements were taken from within 5 °latitude and 25 °longitude of the observation sites (either Zimmerwald, BE, Switzerland, or Ny-Ålesund, Svalbard). Due to the nature of the orbit and observation method, which allows measurements only at sunset and sunrise at fixed local time, the ACE-FTS has relatively few water vapour profiles, and these are clustered close to the beginning and end of the year for Ny-Ålesund, which is located at a very high latitude (78.9° N).

The observations from Zimmerwald, BE, show that already in summer of 2022, water vapour mixing ratios at $0.1\,hPa$ are above preceding years by around 0.5-1 ppmv between day 200 and day 300 of the year (mid July until late October). In 2023, however, unprecedentedly high water vapour mixing ratios were recorded at this pressure level, with almost $9\,ppmv$ being recorded on September 5th 2023 compared to 6.5 - 7.0 ppmv as had been recorded on this date throughout all years between 2015 and 2021.

In Ny-Ålesund, whilst above average water vapour mixing ratios were also present, this is only from summer 2023. It is worth here noting that the observations from MIAWARA-C during 2020 are in some disagreement with the observations from the ACE-FTS and MLS (MIAWARA-C has a positive bias compared to both instruments in this year, compared to previous years where there is a negative bias vs MLS and no bias compared to ACE-FTS). An anomaly similar to that recorded by the MIAWARA-C throughout 2020 then persisted throughout 2024 at 0.1 hPa and 1 hPa, before reducing to near similar levels at the end of 2024.

# 4 Flux Analysis

## 4.1 Longwave down-welling Fluxes

Water vapour strongly influences the Earth's energy budget because it is an effective absorber and emitter of infra-red radiation within specific frequency bands. In the troposphere, water vapor is the most abundant greenhouse gas, and fluctuations in its concentration strongly modulate surface temperatures by altering the radiative fluxes traveling both upwards (outgoing longwave radiation) and downwards (radiation from the atmosphere to the surface). However, the tropospheric lifetime of water vapor is limited by condensation and precipitation processes. As a result, even substantial injections of water vapor into



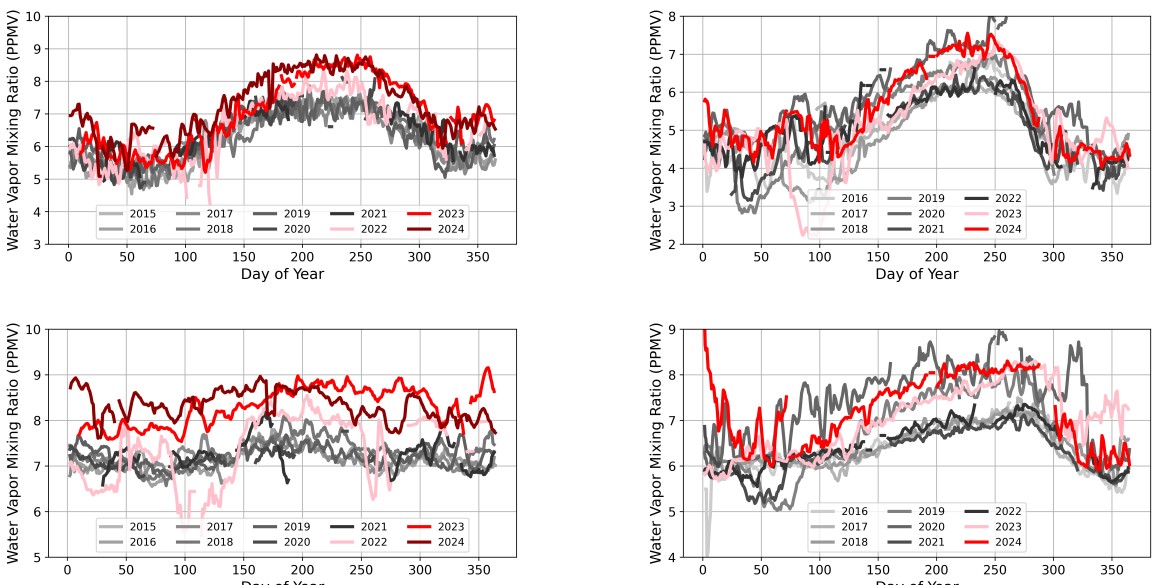

**Figure 2.** Water vapour with day of year, for years between 2015 and 2024 (for MIAWARA; left) and 2018 and 2024 (MIAWARA-C; right) at 0.1 hPa (top row) and 1hPa (bottom row).

the troposphere—such as the one caused by the Hunga volcano eruption—will not persist for long and therefore will have only short-lived effects on tropospheric radiative fluxes.

In contrast, injections of water vapor into the middle atmosphere (the stratosphere and lower mesosphere) can remain there
for much longer, on the order of years to decades, because condensation and precipitation are far less effective at those altitudes. As previously discussed, the Hunga volcano's injection of water vapor into the middle atmosphere could therefore influence the climate for about a decade. This is consistent with findings from Solomon et al. (2010), who demonstrated that a reduction in stratospheric water vapor contributed to slower surface warming (by approximately SI25 %) between 2000 and 2009, implying that any increase in stratospheric water vapor would likely accelerate warming over similar time spans.

It is orthodox in flux calculations for climate studies to calculate changes to downward fluxes at the surface (with the obvious application of surface heating rates). We show below, however, downward flux calculations made at the top of the troposphere, and at the lower measurement response limit of the microwave radiometer observations at 10 hPa. As flux calculations are shown only for single points (as opposed to zonal means or for the entire globe), interannual variability of weather conditions in the troposphere will affect the flux values to a large extent, and the signal coming from the middle atmosphere can be harder
to disentangle from greater or lesser cloud cover, for example. Furthermore, it may be argued that since most down-welling radiation in the thermal infrared wavelengths is absorbed by the troposphere and the surface, affecting the overall heat dynamics and atmospheric stability, the longwave fluxes at the top of the troposphere could be more pivotal in determining global climate patterns than those at the surface.



Monitoring the top-of-atmosphere energy balance is also important for understanding the overall sign and magnitude of
impacts of atmospheric composition change on the net warming or cooling effect on the climate system, as it captures the
energy flow for the whole Earth system, in contrast to downwelling surface flux measurements, which focus on the local signal
at the surface. A brief analysis of the top-of-atmosphere upward longwave flux is thus included in the analysis.

## 4.2   Methods

Radiative transfer simulations are essential for accurately estimating fluxes and heating rates in the atmosphere, but such calcu-
lations often become computationally intensive when carried out at very high spectral resolution or across many atmospheric
profiles. To address this challenge, we compare two approaches— through simulated annealing and lookup tables—which
aim to reduce the computational load without significantly compromising accuracy. By optimising the selection of frequency
points (simulated annealing) and reusing pre-computed absorption coefficients (lookup tables), repeated simulations can be
performed more efficiently for the specific case of examining the impact of composition change on radiative effects.

### 4.2.1   Simulated Annealing Method

In this study, a simulated annealing method is employed to optimise frequency vectors for radiative transfer simulations,
adapting methodologies described by Buehler et al. (2010) and Kirkpatrick et al. (1983). This is initiated with a predefined set
of frequencies that are evenly spaced, from which a series of adjustments of the frequency vector is made to minimise a cost
function $\Lambda$, which evaluates the deviation between simulated irradiance outputs from a high-resolution reference dataset. The
cost function is defined in equation 1, as a function of the integrated value of the absolute value of the difference between the
high-resolution and low-resolution spectra, plus the absolute difference in the integrated spectral intensity between the high-
resolution and low-resolution spectra. In this equation, $\mathbf{I}$ is a cubic spline interpolation function (SciPy Community, 2024). The
cost function was chosen to take into account both the difference in spectral irradiance at each frequency, and the difference in
between the integrated value of this to balance the fact that the spectral distribution of irradiances will change with different
atmospheric profiles and angles which the radiative transfer is calculated at, but also to take into account the fact that the
low-resolution interpolated values would always be negatively biased compared to the high-resolution reference simulation if
the integrated spectra were not considered. The factor of two applied to the first term was found empirically to optimise the
balance between the two terms.

$$\Lambda = 2 \cdot \int_{f_{\min}}^{f_{\max}} |\text{Intensity\_HR}(f) - \text{I}(\text{Intensity\_LR}, f_{\text{LR}}, f)| \, df + \left| \int_{f_{\min}}^{f_{\max}} \text{Intensity\_HR}(f) \, df - \int_{f_{\min}}^{f_{\max}} \text{Intensity\_LR}(f) \, df \right| \quad (1)$$

The process begins by randomly selecting an initial frequency vector and calculating its cost value by comparing the irradi-
ance values provided by the interpolation against the reference spectra. During each iteration of the annealing process, a new
candidate set of frequencies is generated by slightly modifying the current set. The new set comprises the old set, with a single
frequency replaced by another. The replacement frequency is either one of the 30 frequencies with the maximum difference




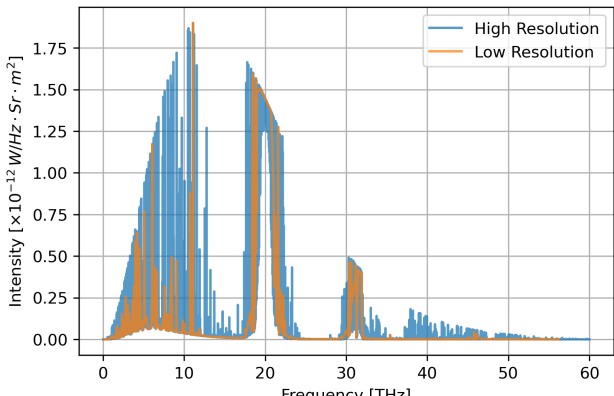

**Figure 3.** Spectral Intensity of the high-resolution- containing 4000 evenly spaced frequency points- and the low resolution- containing 400 frequency points- spectra at a hight of 30km and a zenith angle of $30°$

.

when the interpolated set compared to the reference set, or a random frequency in the set of reference frequencies that is not
currently used (a $50\,\%$ probability is used to decide which, on each iteration). This new set is evaluated, and if it shows improved or equivalent performance- guided by a probabilistic acceptance based on the simulated annealing temperature- it is accepted as the new solution. The temperature parameter decreases with each iteration, reducing the probability of accepting sub-optimal solutions as the process progresses, thereby refining the selection towards a globally optimised set of frequencies.

    The frequencies used for the radiative transfer analysis are depicted in Figure 3. While the low-resolution simulation lacks
many of the finer spectral features, the integrated intensity between the high- and low-resolution spectra shows no significant difference at 18 km asl. This indicates that the selected frequency vector adequately represents the spectrum in question.

    The goal of this optimisation is to enhance the computational efficiency and accuracy of radiative transfer simulations. This approach uses the ARTS radiative transfer simulator to simulate outputs at 19 angles between the zenith and the horizon. Pencil beam calculations are made for the 400 frequencies previously selected for their representation of the total flux. The total flux
is then found by integrating over all frequencies and 15 elevation angles between $0°$ (zenith) and $90°$ (horizon).

### 4.2.2   Lookup Table Method

Radiative transfer calculations typically involve determining the absorption coefficients for each absorbing species, across various pressure levels, and for each atmospheric path. This process can be computationally intensive and time-consuming, es-pecially when performed repeatedly across many atmospheric profiles. To streamline these calculations, one effective approach
is to use a lookup table. Instead of recalculating the absorption coefficients for every new simulation, a lookup table is created and populated with data calculated at specific frequencies beforehand. This table includes pre-computed values for different conditions and parameters relevant to radiative transfer, such as gas concentrations, pressure levels, and temperatures.





Once established, this lookup table can be quickly referenced in subsequent model runs. By doing so, the model bypasses the need to perform detailed absorption coefficient calculations from scratch each time, thus substantially reducing the compu-

tational load. This approach can lead to significant savings in computational time, particularly if the frequency vector remains unchanged across all simulations. A python wrapper to perform fast flux calculations to work with ARTS has recently been developed (pya, 2024), which allows the set-up of flux calculations using the efficient lookup table method. It also takes advantage of recent developments to ARTS, which has expanded the frequency range of possible radiative transfer simulations from microwave and infra-red bands to also include visible and ultra-violet bands (Brath et al., 2024).

### 4.3   Method Comparison


To verify the radiative transfer calculations, long-wave down-welling fluxes are simulated from a recent climatology (2016 - 2021) of water vapour profiles retrieved with the MIAWARA radiometer. The same is done with the water vapour profiles retrieved between 2022 and 2024 with the daily water vapour profiles retrieved from the instrument. By performing simulations with both the lookup table and the full ARTS method, the calculated fluxes can be compared, and an error estimate of the model

assumptions can be made.

As may be seen from figure 4, there is a small difference between the absolute values of fluxes that are calculated by the two methods. Although the lookup table method calculates stronger (more negative) fluxes between winter and mid-summer, stronger fluxes are calculated by the full ARTS method in the late summer and Autumn, coinciding with the largest water vapour anomalies observed throughout the period shown. The mean difference between the two methods is $0.075\,\mathrm{W\,m^{-2}}$, with

the lookup table method calculating stronger fluxes, and the standard deviation of $0.077\,\mathrm{W\,m^{-2}}$, or $0.4\,\%$.

Despite this, the anomalies predicted by both methods, by comparing fluxes simulated from the climatology to fluxes calculated in the post-eruption period, a very good agreement is found. Throughout the period, the entire ARTS method showed more intense fluxes by a mean of $0.0017\,\mathrm{W\,m^{-2}}$, with the standard deviation of the differences being $0.003\,\mathrm{W\,m^{-2}}$ or $0.2\,\%$

The simulated annealing method produces an increase in the down-welling flux by $0.016\,\mathrm{W\,m^{-2}}$ between January 2016 and

March 2024, whilst the lookup table method calculates an increase by $0.014\,\mathrm{W\,m^{-2}}$, corresponding to a $0.1\,\%$ increase in total down-welling flux at this height. Due to the faster computation time, the lookup table method was used for further calculations presented.

### 4.4   Shortwave Radiative Impact

As highlighted in section 1, short-wave fluxes need to also be considered when making assessments of total radiative effects

of changes in atmospheric composition, i.e. how radiation coming from the sun is attenuated and induces atmospheric heating. Previous studies analysing shortwave flux effects from Hunga-induced atmospheric composition changes have focused on the impact of increased aerosol concentrations on solar fluxes; either being absorbed or scattered, having the result that short-wave fluxes at the surface were reduced in the period following the Hunga eruption (Sicard et al., 2025). Another impact is caused directly by the increase in water vapour: water vapour molecules scatter and absorb light in the short-wave part of the spectrum,

allowing changes in the concentration of this gas to impact the down-welling fluxes and induce local heating effects.



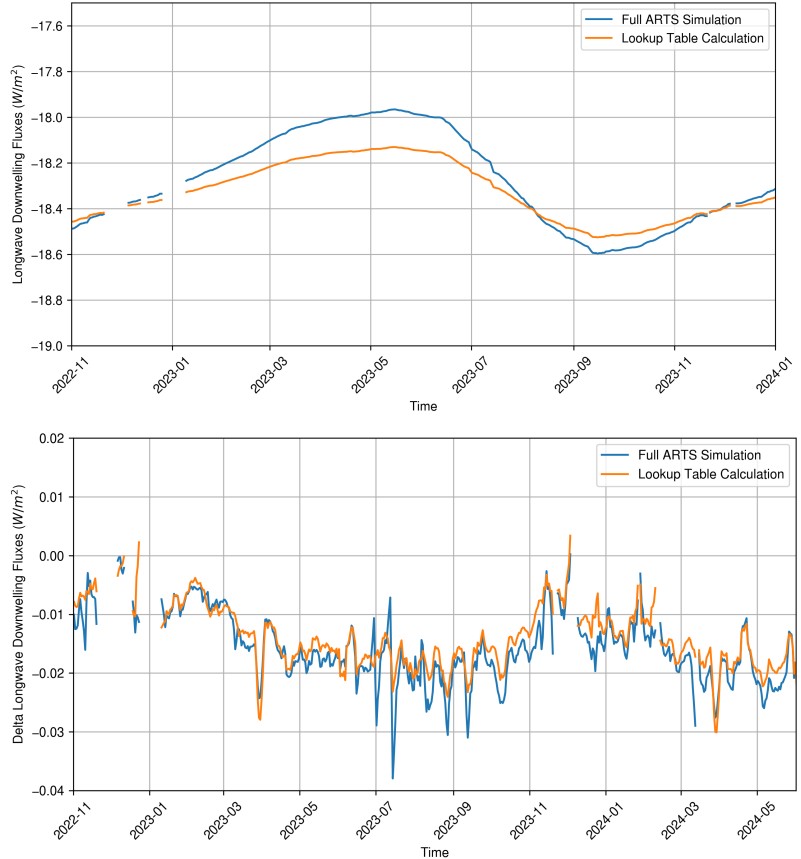

**Figure 4.** The long-wave down-welling flux anomaly above Zimmerwald, Bern, at 14 km, simulated from atmospheric profiles using the full radiative transfer code, and the lookup table method. The negative sign corresponds to a positive flux in a downwards direction, hence a negative anomaly represents a stronger downward flux.

The propagation of solar fluxes is a function of the zenith angle of the sun relative to the observer's position, and thus are a function of the time of day, but also highly seasonal dependent at high (arctic) and low antarctic latitudes.

A computation of change in the mean down-welling short wave fluxes was also calculated using the lookup table method. To calculate the impact that the changes in water vapour had on fluxes at the tropopause, the radiative transfer code is run at wavelengths of $300\,\mathrm{nm}$ to $5000\,\mathrm{nm}(60\,\mathrm{THz}$ to $1000\,\mathrm{THz})$ at a wavelength resolution of $2.35\,\mathrm{nm}$. The included species are $H_2O$, $O_2$, $CO_2$, $CO_4$, and $O_3$. The solar flux entering at the top of the atmosphere is calculated for a fixed earth-sun distance of $1.495\,978\,707 \times 10^9\,\mathrm{km}$, and the solar spectrum set as that recorded in May 2004 Coddington et al. (2017). Both absorption and scattering of atmospheric species are considered, but not emission.

The relative position of the sun to the surface is calculated using the PyEphem python library (Rhodes, 2023). From this package, the right ascension and declination of the sun are found from a precise date and time, from which the solar zenith





longitude and latitude (the Earth's longitude and latitude at which the sun is directly overhead) are calculated. These serve as inputs to the lookup table flux calculation package, and so combined with an atmospheric profile allow the calculation of down-welling short-wave fluxes.

As with the long-wave calculations, a standard atmospheric profile of all other atmospheric constituents is used, with a
climatological temperature profile, and only the water vapour profile taken from measurements. For the climatological flux calculation, the average water vapour data from 2016 to 2021, based on the day of the year, is used. To accurately compare solar fluxes, the sun's position must match exactly. Therefore, in this study, the climatology was calculated based on the sun's position during the post-Hunga eruption period, specifically between 2022 and 2024. To estimate the daily mean solar down-welling flux, calculations were performed for each hour when the solar zenith angle was less than 95 degrees (i.e. when the sun
was above the horizon, or less than $5°$ below the horizon). Then these results were interpolated to the minute resolution using a cubic spline method (SciPy Community, 2024). Finally, the daily mean down-welling flux was calculated as the average of these interpolated values.

## 4.5 Flux Analysis from Zimmerwald and Ny-Ålesund

One key difference in the retrieval technique between the measurements from Zimmerwald and Ny-Ålesund is that whilst
those from Zimmerwald have a static a priori water vapour profile (taken from a climatological profile which has its source from ECMWF IFS analysis and MLS observations), the retrievals from Ny-Ålesund takes the lower a priori profile from the ECMWF analysis valid closest to the MIAWARA-C observation time. This means that below the lower measurement response limit of the MIAWARA (typically around 30 km or 10 hPa), water vapour profiles may deviate from the climatology even if there is a significant error in the a priori profile. For this reason, downwelling fluxes were calculated again at 10 hPa at
Zimmerwald and Ny-Ålesund.

The anomalies presented in figure 5 show a positive response (meaning less downwelling flux) for solar flux and a negative response for infrared fluxes for almost all the period analysed after 2023 for both locations. The reduction in shortwave flux has a seasonal dependence due to the difference in the top of atmosphere downwelling solar flux throughout the year, as well as the observations, presented having the general characteristic of a greater water vapour mixing ratio anomaly in summer compared
to winter. The seasonal cycle is even more profound in Ny-Ålesund (at $79°$ North) compared to Zimmerwald (at $47°$ North), where the polar night lasts for four months.

The mean downwelling anomaly at this pressure level in Zimmerwald was $0.20\,\mathrm{W\,m^{-2}}$ ( $0.22\,\mathrm{W\,m^{-2}}$ from longwave only) for 2023 and $0.19\,\mathrm{W\,m^{-2}}$ ( $0.21\,\mathrm{W\,m^{-2}}$ from longwave only) for 2024. Above Ny-Ålesund, this was $0.16\,\mathrm{W\,m^{-2}}$ ( $0.17\,\mathrm{W\,m^{-2}}$ from longwave only) for 2023 and $0.23\,\mathrm{W\,m^{-2}}$ ( $0.24\,\mathrm{W\,m^{-2}}$ from longwave only). This demonstrates therefore
a slight increase in the total downward radiative fluxes coming from the mid-stratosphere during 2023 and 2024 at both locations as a result of the Hunga-induced water vapour increase.

For completeness, the top of atmosphere upwards fluxes are also analysed. In contrast to the downward flux measurements, part of the atmosphere cannot be simply excluded from these measurements to avoid the previously discussed impact of the measurement a priori profile on the upward fluxes. As is shown in Figure 6, the impact of the water vapour increase on the top

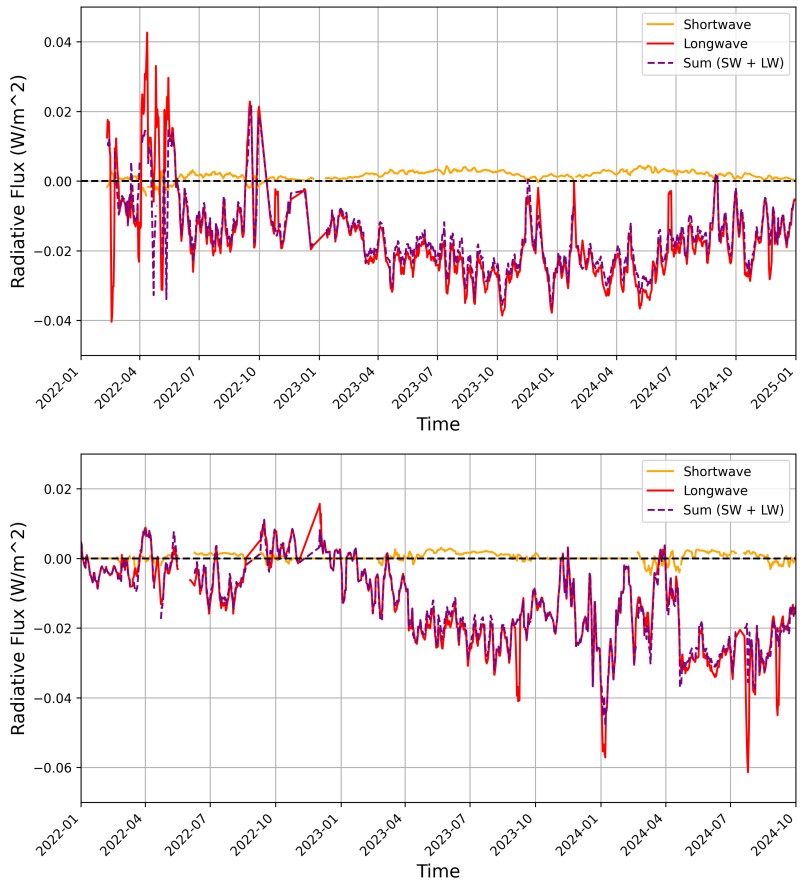

**Figure 5.** The short-wave and long-wave down-welling fluxes at 10hPa above the site of Zimmerwald (top) and Ny-Ålesund (bottom).

of atmosphere upward fluxes is positive at Zimmerwald, indicating a cooling effect on the climate as a whole, and negative at Ny-Ålesund. The reason for the increase in up-welling long-wave flux is here likely due to the altitude of the maximum water vapour anomaly- i.e. in the upper stratosphere and lower mesosphere. As the thermal gradient is positive in the stratosphere, the additional water vapour at those higher altitudes is effectively radiating more thermal energy out to space, thus increasing the outgoing long-wave flux. By contrast, at Ny-Ålesund, the bulk of the water vapour increase extends to lower altitudes (and cooler stratospheric conditions), thereby inhibiting the net emission of energy to space. As a result, the top-of-atmosphere flux anomaly becomes negative there, implying less energy is lost to space and thus a potential warming effect.

### 4.6 Local Heating Rate Anomaly

Whilst the increasing opacity caused by increases in water vapour results in long-wave emission, the local effect is one of cooling, assuming that all other effects remain constant. The lookup table method, described above and used to calculate



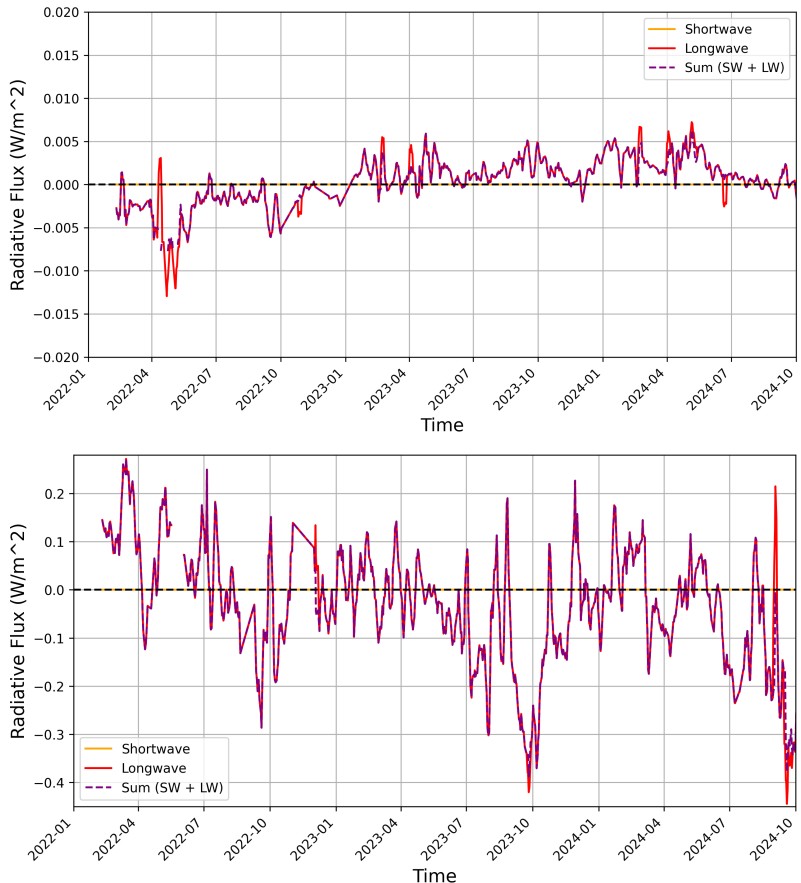

**Figure 6.** The short-wave and long-wave up-welling upwelling fluxes at the top of the atmosphere above the site of Zimmerwald (top) and Ny-Ålesund (bottom).

down-welling long-wave fluxes, was also used here to calculate the heating rates in the atmosphere, using the climatology that formed the basis previously, as well as with atmospheric profiles using the measured water vapour profiles as performed above.

The mesosphere exhibits an atmospheric circulation from the summer pole to the winter pole, as air rises from the summer pole, and is transported horizontally at mesospheric altitudes and descends over the winter pole, which is often referred to as residual circulation (Lindzen, 1981; Smith, 2012; Becker, 2012). Temperature anomalies can affect this flow by affecting temperature gradients. However, temperatures are also affected directly by the strength of the circulation due to the vertical and horizontal potential temperature gradients in the middle atmosphere. In Yu et al. (2023), results are presented showing that in August 2022, southern hemispheric mesosphere cooling rates due to increased long-wave radiation and due to changes in dynamics were of the same amplitude.

The heating rate profile throughout 2023 and 2024 reveals that during these years, there was increased water vapour-induced radiative cooling compared to the 2016-2022 climatology at pressures below $10\,\mathrm{hPa}$ on average for both Zimmerwald and



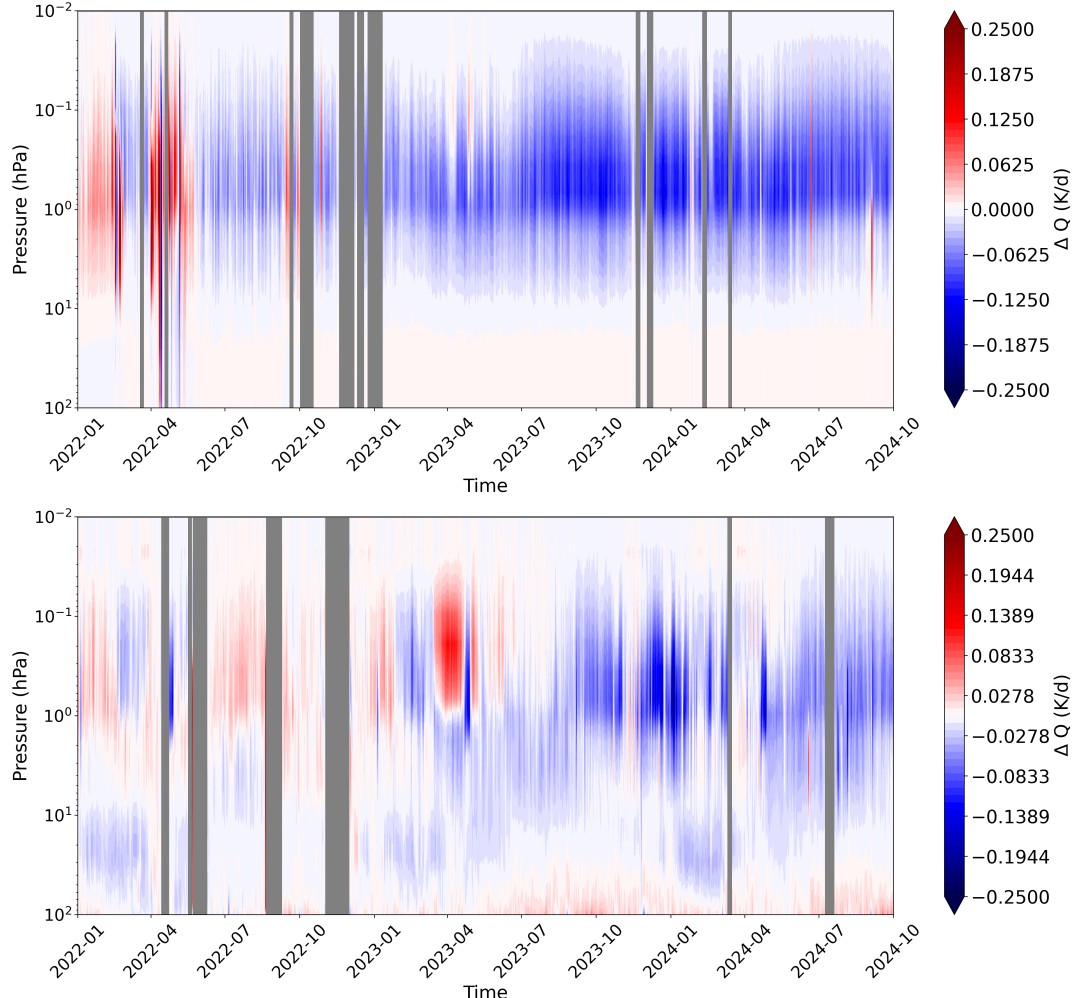

**Figure 7.** Heating rate anomaly profiles from long wave emission and absorption, calculated above Bern, Switzerland (top) and Ny-Ålesund, Svalbard (bottom) between 2022 and 2024.

Ny-Ålesund. At Zimmerwald, the heating rate anomaly at $0.8\,\mathrm{hPa}$ remained between $0.10\,\mathrm{W\,m^{-2}}$ to $0.125\,\mathrm{W\,m^{-2}}$ between August and November 2023. From the 1st January 2023 until 1st November 2024, whilst the mean heating rate anomaly was $-0.09\,\mathrm{K\,d^{-1}}$ at $1\,\mathrm{hPa}$ and $-0.05\,\mathrm{K\,d^{-1}}$ at $0.1\,\mathrm{hPa}$. In Ny-Ålesund, at the start of 2023, there is a notable positive heating rate anomaly in the mesosphere, during a mesospheric dry phase. After April 2023, the effects of the Hunga water vapour injection started to show here, and a negative heating rate anomaly persists in the stratosphere and mesosphere almost without interruption.

Shortwave absorption by water vapour molecules is substantially weaker than their longwave absorption. Nevertheless, to fully assess the total radiative heating rate anomaly, it remains important to include these shortwave contributions alongside





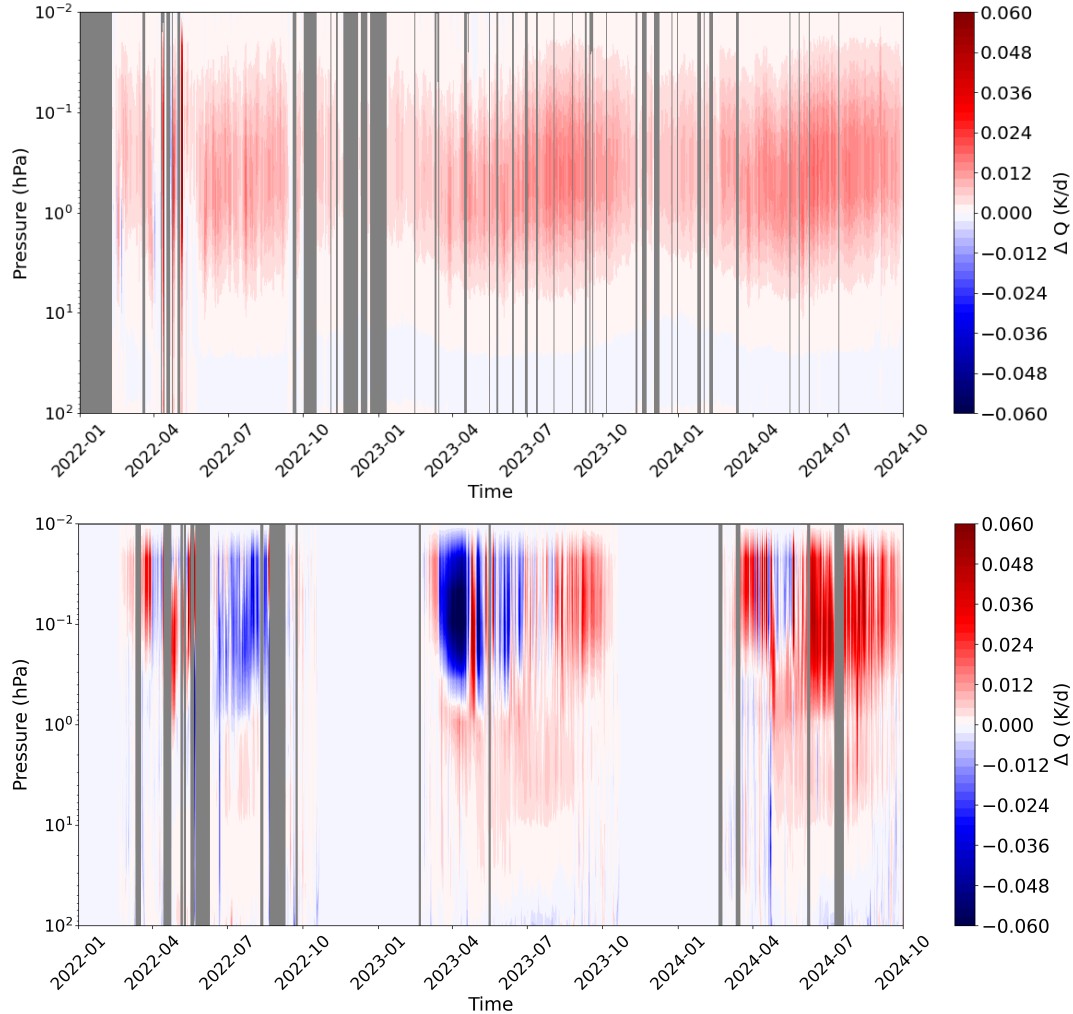

**Figure 8.** The heating rate anomaly after the Hunga eruption as a result of the increased short-wave absorption above Zimmerwald (top) and Ny-Ålesund (bottom).

the longwave heating rates. Because water vapour does not significantly emit radiation in the shortwave at middle-atmospheric

345 temperatures, emission was not considered by the radiative transfer model.

From Figure 8, a consistent warming effect of approximately $0.006\,\mathrm{K\,d^{-1}}$ is evident at 1 hPa over Bern. In contrast, a much stronger seasonal cycle is observed in Ny-Ålesund. No heating rates are calculated during the four-month polar night, but around the summer solstice, the relatively large mean daily solar flux at the top of the atmosphere leads to pronounced heating anomalies. Of particular note is the negative heating anomaly in spring 2023 between 0.4 hPa and 0.01 hPa, followed by a

350 positive anomaly increasing through summer 2024 at similar pressure levels.



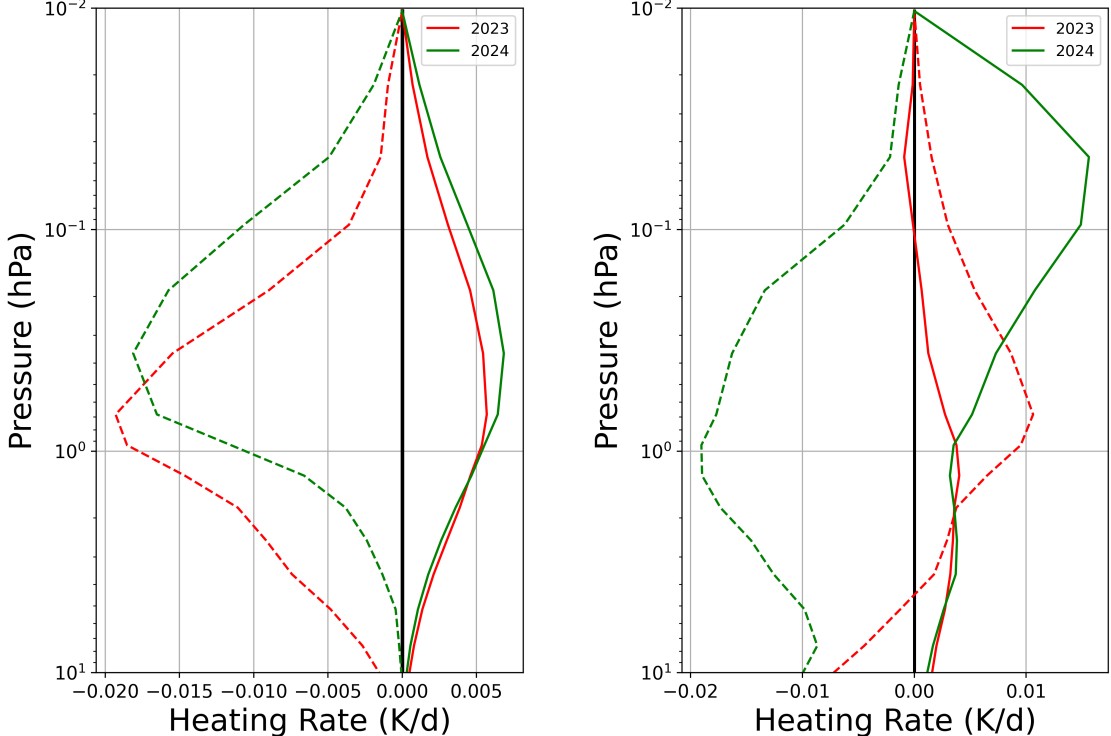

**Figure 9.** Mean heating rate anomalies for Zimmerwald for the entire year (left) and Ny-Ålesund between mid-February and late October (right), with the anomaly caused through changes in longwave radiances in dashed lines, and shortwave radiances in solid lines.

In Figure 9, the combined effect of longwave and shortwave heating is combined, for whole years above Zimmerwald, and for the outside the polar night (day of year 51 until 287) at Ny-Ålesund. The results show a conclusive net cooling over both years at Zimmerwald, across all altitudes analysed, with a shortwave heating approximately $25\,\%$ of the longwave cooling impact, with the height of the maximum heating anomaly increasing from 2023 to 2024. In Ny-Ålesund, there is a slightly different picture, with no significant longwave cooling in 2023, and a larger mean impact from shortwave heating that year. In 2024, when there is a more decisive water vapour anomaly, the longwave heating rate is more significant, at $-0.019\,\mathrm{K\,d^{-1}}$ at 1 hPa. However, above 0.15 hPa, the shortwave heating dominates the longwave cooling, and the net impact is positive, with a mean anomaly of $0.01\,\mathrm{K\,d^{-1}}$ for the non-polar night period.

## 5 Conclusions

Whilst uncertainty persists about how long the anomaly of water vapour in the middle atmosphere will persist, and the global distribution of this increase in water vapour, it is generally agreed that higher than average total middle atmosphere water vapour will persist at least until 2027, with the potential for significant water vapour to remain until 2032 or longer.



The radiative impact studies conducted in this paper have found that the increase in long-wave down-welling fluxes at 10 hPa has outweighed the decrease in short-wave down-welling fluxes by approximately a factor of 10. The climatic impact of this at the surface is admittedly very limited. Whilst the increase in long-wave down-welling flux increased by between $0.010\,\mathrm{W\,m^{-2}}$ to $0.04\,\mathrm{W\,m^{-2}}$ for over $90\,\%$ of the period between March 2023 and May 2025 at Zimmerwald, this is approximately $1\,\%$ of the change due to the increase in carbon dioxide since pre-industrial times ($1.63\,\mathrm{W\,m^{-2}}$ to $2.01\,\mathrm{W\,m^{-2}}$ increase between 1750 and 2011) (Myhre et al., 2013).

A more substantial climatic influence of the Hunga-induced water vapour enhancement may occur through local heating-rate changes within the middle atmosphere. Between 1 January 2023 and 1 January 2025, we find net (long-wave plus short-wave) heating-rate anomalies of about $-0.012\,\mathrm{K\,d^{-1}}$ at $1\,\mathrm{hPa}$ above Zimmerwald and $-0.016\,\mathrm{K\,d^{-1}}$ at $1\,\mathrm{hPa}$ above Ny Alesund, which when sustained over multiple months has the potential to alter middle-atmospheric temperature gradients and winds. Such dynamical shifts can propagate downward, affecting tropospheric circulation and weather patterns. Additionally, these temperature changes could influence stratospheric ozone chemistry, with possible consequences for both climate regulation and surface-level UV radiation.

*Data availability.* MIAWARA data can be accessed through the landing page at: doi.org/10.60897/pdyc-8v84 and MIAWARA-C can be accessed through the landing page at doi.org/10.60897/gk0h-3b75.

*Author contributions.* AB ran radiative transfer codes and performed all data analysis with input and suggestions from GS. The manuscript was written by AB, with edits from GS and AM.

*Financial support.* This work had been completed as part of the "Swiss H2O Hub for High-quality water vapor measurements from ground to space" project financed by GCOS Switzerland.

*Competing interests.* Gunter Stober is an editor at Annales Geophysicae.

*Acknowledgements.* The authors thank Manfred Brath and the other ARTS developers for their continued work with the radiative transfer software and user support. The authors are also members of the Oeschger Centre for Climate Change Research (OCCR). The Atmospheric Chemistry Experiment (ACE), also known as SCISAT, is a Canadian-led mission mainly supported by the Canadian Space Agency.



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
