# Peer review of "Radiative impact of increased middle atmospheric water vapour in the aftermath of the Hunga 2022 volcanic eruption at two locations in the Northern Hemisphere"

_EGUsphere, 2025_

## Author Comment (AC2)

Thanks for the comments from both reviewers for the time taken to have performed the review of our paper. Their detailed commends have helped us improved the clarity, structure and scientific rigor of the article. In response to the major concerns raised by both reviewers, we have undertaken substantial revisions to address the key issues identified:

**Comment**: Important information is missing: retrieval altitude range (i.e., range sensitive to H2O), vertical resolution of the profiles, degrees of freedom. It would also be good to show sample averaging kernels of the H2O retrieval.

**Response**: All suggested changes implemented

**Change**: Added [line 98]: "The current retrieval framework gives a temporal resolution of 1 day$^{-1}$, and the measurements have a vertical resolution ranging from 15 km (at 30 km) to 20 km (at 70 km). The median degrees of freedom for signal obtained from MIAWARA retrievals during 2024 was 1.94. A representative averaging kernel from the instrument is shown in Figure 1."

Added [line 117]: Measurements in the current retrieval framework have a temporal resolution of 1 day$^{-1}$ and a vertical resolution ranging from 13 km at 30 km asl to 20 km at 70 km asl. The median degrees of freedom for signal obtained from MIAWARA-C retrievals during 2024 was 3.86. A representative averaging kernel from the instrument is shown in Figure 1.

**Comment**: Retrieval errors are not discussed at all. How do the H2O retrieval errors translate to flux errors and errors in the heating rates. Right now, the reader does not know, whether the heating rates are larger than the corresponding errors.

**Response**: This was indeed neglected in the original manuscript. In order to qualify the errors without having to run the radiative transfer simulations again, a compromise was drawn, where the mean heating rate/flux error would be calculated from a mean water vapour profile and mean errors. Although the heating rate and flux response to changes in water vapour is indeed nonlinear, this approach was thought sufficient to give a first order approximation of the errors for a single profile. It was found that although the heating rates changes a single profile are not significant (outside the range of uncertainty), when the mean over the years 2023 and 2024, the heating rate differences do indeed exceed the uncertainty estimates by a factor of (depending on which location and instrument) around 10.

**Change**: Added mean retrieval error at two heights in instrument description for MIAWARA and MIAWARA-C

**Added**:

Uncertainty Assessment

To assess the uncertainties in the presented heating rate calculations, several sources of uncertainty need to be considered. The first and primary source is the retrieval error, which arises from both measurement error and inversion smoothing. The second source of uncertainty comes from demonstrating that the changes to the heating rate exceed the natural variability of the middle-atmospheric water vapour state. The third source is the error originating from the radiative transfer model itself. To demonstrate statistical significance, it is required that the Hunga-induced heatingrate changes exceed twice the total standard error, corresponding to a confidence interval of approximately 95 %.

To account for retrieval errors in the heating rate estimations, several assumptions were made. First, although the retrieved profiles have a high vertical correlation due to the shape of the averaging kernels (for radiometric observations there is never perfect sensitivity at each retrieval level, and instead profiles are somewhat smoothed), it was assumed that there was no error correlation between successive measurements. Second, it was assumed that the heating rate error for each profile could be approximated from the difference between a mean water vapour profile plus the mean retrieval error, and the mean water vapour profile minus the retrieval error.

To this end, radiative transfer simulations were made by repeating the methodology previously outlined in Section 2.4.2 for a mean water vapour profile, a mean plus water vapour retrieval error profile, and a mean minus water vapour retrieval error profile. The maximum heating rate error due to measurement error occurs at pressures between 0.5 hPa and 1 hPa. This is despite the measurement error being lower at these pressure levels than in the upper mesosphere and mid- to lower stratosphere, due to better measurement response at this range. Because the heat capacity of very thin air is smaller, a fixed radiative perturbation translates into a larger temperature tendency at lower pressure levels. Consequently, a given change in water vapour has a greater effect on heating rate in this pressure range than in the mid- to lower stratosphere.

Similarly, the component of standard error due to the natural variability of water vapour in the middle atmosphere was determined by finding the standard deviation of water vapour anomalies (post-Hunga minus 2015–2021 climatology) at each pressure level, and calculating the heating-rate perturbation for the mean water vapour profile perturbed by these quantities.

To estimate the standard error arising from radiative transfer uncertainties, the comparison of the two model applications—simulated annealing frequency vector and lookup table—for calculating surface downwelling fluxes, as described in Section 2.4.2, was used. This comparison shows a mean difference of about 0.5 %, which, although both methods rely on the same radiative transfer model, line catalogue, and line-shape parameters, provides a practical estimate of the uncertainty introduced by the lookup-table methodology.

With these uncertainties in hand, the effective number of independent observations is used to calculate the yearly mean heating-rate shift from the climatology to the post-Hunga years 2023 and 2024 according to equation 2, where $\sigma_g^2$ is the variance of daily geophysical variability, $\sigma_m^2$ is the retrieval variance, $N_{eff}$ is the autocorrelation-corrected number of independent days, $\sigma_{RT}^2$ is the systematic 0.5 % radiative transfer error, and SE is the resulting standard error of the annual mean shift.

$$SE = \sqrt{[\ (\sigma_g^2 + \sigma_m^2)\ /\ N_{eff} + \sigma_{RT}^2\ ]}$$

Similarly, to estimate the error in a single shortwave and longwave flux simulation, a perturbation equal to the retrieval error was added to the mean water vapour profile, and a radiative transfer simulation was run. This was followed by a run with the mean profile, and another with the mean profile minus the retrieval errors. This led to an uncertainty of 0.37 W m⁻² and 0.02 W m⁻² over Zimmerwald for the longwave and shortwave simulations, respectively. At Ny-Ålesund, the single-profile error was 0.37 W m⁻² and 0.02 W m⁻² for longwave and shortwave, respectively. When a

yearly error calculation was made again using the above equation, the yearly error associated with the flux calculations was obtained.

**Comment**: There are significant biases and discontinuities in the MIAWARA-C time series that should be addressed and understood before the dataset is used for scientific analyses.

**Response**: This was investigated in the original data analysis of the water vapour time series for the MIAWARA-C. I think here the reviewer is referring to figure 2 (bottom right panel). Notably, at the beginning of years 2016 and 2024 there are large oscillations that do not seem to fit with the rest of the time series. The jump at the beginning of 2024 appears to be larger than the rest of the time series for several reasons. The first is of the way in which the data is presented. The presented lines in fact do not show the daily data, but a 3 day moving average (this is now stated explicitly in the manuscript). The way this was employed, however, did not require a minimum number of observations in the 3 day moving window. In the case of the start of 2024, there was no converged retrieval for the first day of 2024. This meant that the first day of 2024 is treated as a daily observation and appears as a spike, due to the also very high water vapour mixing ratios in this period. When the two dimensional (time vs pressure) data is seen, the data appears vastly less discontinuous, see the figure below.

[Figure]

**Change**: Figure 2 has now been changed so that two points are required for a moving average. It has been added to the figure description explicitly that this is a three day moving average.

**Comment**: The net radiative fluxes should be defined early in the paper (downward – upward (my recommendation) or the other way around). This is not done, but you rather speak of upward and downward fluxes (which can in principle be negative and positive), which makes it more difficult to follow the descriptions.

**Response**: This is now defined in section 2.4.

**Change**: [line 12] In this study, down-welling fluxes are assigned a negative sign  and up-welling

fluxes a positive sign . Accordingly, an increase in downward flux appears as a more negative value, whereas an increase in upward flux appears as a more positive value.

**Comment**: Line 13: "Matoza et al. (2022)", "(Basha et al., 2023)" Line 90: "Buehler et al. (2018)"

Wrong cite command.

**Change**: corrected

**Comment**: Line 22: "The water vapour in the middle atmosphere is relatively stable because of the few sinks in this part of the atmosphere."

I suggest mentioning the sinks here.

**Change**: Added: In the stratosphere, water vapour is mainly supplied by the upward transport of tropospheric moisture across the tropical tropopause, and at higher altitudes predominantly by the oxidation of methane advected from below. Major sinks include photolytic decomposition, as well as removal through condensation onto polar stratospheric and mesospheric ice particles, which is followed by gravitational sedimentation.

**Comment**: Line 85: "A balancing calibration scheme using a reference view that optimizes noise and linearity (Forkman et al., 2003)."

Sentence is incomplete.

**Change**: [added: line 89] To minimise noise and optimise the linearity of the calibrated spectra, a balancing calibration scheme using a reference view is employed (Forkman et al., 2003).

**Comment**: Lines 90 and 104: How are the covariance matrices chosen, both the a priori and the measurement covariance matrix? How many degrees of freedom are there? What is the vertical resolution of the profiles and the altitude range sensitive to H2O. Please provide more information on the retrievals.

**Response**: The requested infromation about the degrees of freedom and vertical resolution has now been added to the instrumental sections to give a better idea about each instrument's performance. The details about the choice of error covariance matrices and a priori profile for the MIAWARA retrievals are discussed in detail in the publication by Bell et al. (2025), and it is the authors' opinion that this discussion does not need to be repeated here. For MIAWARA-C, the same approach is taken with one subtle difference: that the lower part of the a-priori profile is taken from the ECMWF analysis valid at the time of the retrieval.

**Change**:
Addeed : [line 97] The choice of instrumental and a-priori error covariance matrices is also described in detail in Bell (2025).
Added [line 98]: "The current retrieval framework provides a temporal resolution of 1 day$^{-1}$, and the measurements have a vertical resolution ranging from 15 km (at 30 km) to 20 km (at 70 km). The median degrees of freedom for signal obtained from MIAWARA retrievals during 2024 was 1.94. A representative averaging kernel from the instrument is shown in Figure 1."

Added [line 116]: Measurements in the current retrieval framework have a temporal resolution of 1 day$^{-1}$ and a vertical resolution ranging from 13 km at 30 km asl to 20 km at 70 km asl. The median

degrees of freedom for signal obtained from MIAWARA-C retrievals during 2024 was 3.86. A representative averaging kernel from the instrument is shown in Figure 1.

Added [line 114]: The retrieval process is similar to that explained in Bell (2025), with the same handling of measurement and a priori error covariance matrices, but has one key difference in that a priori water vapour profiles for the lower stratosphere contain information from the ECMWF forecasts corresponding to the retrieval time.

**Comment**: Line 108: "the key advantage that is offered when compared to ground-based instrumentation is the horizontal coverage of measurements."

And the typically much better vertical resolution, right?

**Response**: Correct

**Added** [line 124]: and sometimes offer better vertical coverage as well.

**Comment**: Line 132: "Much has been written about the evolution of the initial water vapour plume (Schoeberl et al., 2023a; Nedoluha et al., 2024)."

I suggest to have a look at the discussion in Niemeier et al., (2023), dealing with MLS observations and ICON simulations (https://agupubs.onlinelibrary.wiley.com/doi/10.1029/2023GL106482), including a nice description of the reasons for the evolution of the H2O plume.

**Response**: Thanks for the paper recommendation. I have added it as a citation.

**Comment**:Line 134: "Within a month, there was a significant northward movement of the plume across the equator"

But not much further (& it would be good to mention the latitude of the volcano)

**Modified** [line 37]:Several months after the eruption, a portion of the water vapour plume had been transported just across the equator into the northern hemisphere (from the eruption latitude of 20.5° S), which has been proposed to have been partly due to equatorial Rossby waves generated by long-wave cooling of stratospheric air by the increased water vapour itself (Schoeberl et al., 2023a)

**Comment:** Line 134: "which was attributed to infrared cooling effects associated with the high levels of water vapour"

The strong initial descend of the H2O plume was explained by the radiative cooling by H2O, but I'm not sure about the meridional transport? I may be wrong.

**Response:** According to the Scoeberl et al. (2023) paper, the cross-equatorial transport is attributed to the (radiative) cooling-forced Rossby gyres. The Niemeier et al. (2023) paper that you recommended also seems to suggest that the cooling is important for the transport ["According to our passive tracer simulations, the equatorial crossing is enhanced by the radiative cooling of the H2O cloud (Figures S2a and S2d in Supporting Information S1) and does not occur in the simulations with a different QBO phase (Figures S2g and S2j in Supporting Information S1)."]

**Comment:Line 141:** "With MLS data, the transport of the water vapour anomaly is visible to as far north as 80."

Not before the beginning of 2023, right? Perhaps it would be good to mention briefly, when the H2O plume reached northern mid-latitudes.

**Response:** Actually in the paper by Nedoluha et al. (2023), water vapour at Table mountain in California (39.5 N) is found to already be above climatology at 50km (by around 0.5 ppmv) and 70km (by 0.5-1.0 ppmv) in mid 2022. This is also the case in Bern. They credit this early rise to increased methane oxidation due to slower ascent rates, however. We cannot rule out that a portion of this small increase was from the Hunga plume (see supplementary video) of this article.

**Modified:** With MLS data, the transport of the water vapour anomaly is visible to as far north as 80°N (in late 2023), and earlier in 2023 at mid-latitudes.

**Comment**: Line 143: "(see Supplementary Material)."

There doesn't seem to be a supplement?

**Response:** Appologies if this is my fault. I will resolve this.

**Comment:** Line 147: "Water vapour measurements above Ny-Ålesund exhibit for the MIAWARA-C and ACE-FTS slightly lower values compared to MLS retrievals,"

Not for 2019 and 2020! What is going on in these years?

**Response:** In 2019 and 2020 the bias of MIAWARA-C with respect to the Aura-MLS appears to switch sign, and record anomalies (up to this time) are recorded by the MIAWARA-C. This appears at first to be an instrumental artifact, and so this was investigated.

The only instrumental problem that fits with this timeline, was that a window on the instrument had broken, potentially allowing more moisture than normal to enter the instrument, and, though this could be responsible for larger noise in the measurements due to potentially greater attenuation of the middle atmospheric signal, we see no reason for this to cause the MIAWARA-C minus Aura-MLS bias to switch from being negative to positive.

In contrast to the theory that this is an instrumental problem, one author of this paper is currently investigating an anomaly in wind observations from several locations across northen europe. It can be seen in the plots in 2019 and 2020, the spring transition from westerlies to easterlies appears to be disrupted. More results will be presented outlining a theory for these anomalous results.

We would like to say that the results for MIAWARA-C, though disagreeing with the Aura-MLS for this time period, are trusted for this study.

[Figure]

*Meridional and zonal winds in the MLT region from Esrange, Sweden, and Alta, Norway.*

**Comment:** Figure 1: what about the different vertical resolutions of the measurements? Were the satellite profiles convolved with the AVK of the MW retrievals? Also: why don't you show profiles?

**Response:** In Fig. 1 we are examining the broad Hunga Tonga water-vapour enhancement whose vertical extent is ~10 km. Degrading the satellite profiles by 3–4 km more (i.e. convolving with the MW kernels) would change individual points by < 0.2 ppmv (< 2 %). For the ACE-FTS, due to the sparsity of observations, they are selected within a 5 ° latitude and 25 ° longitude of the observation sites, meaning that the horizontal variability is likely to outweigh the differences made by convolving the measurements. A colormap of a comparison of three instruments is not shown due to the difficulty of comparing three three-dimensional plots, vs comparing three dot or line plots on the same axes.

**Comment:** Figure 1, right panel: what is the reason for the MIAWARA-C high biases in 2019 and 2020. They are partly larger than the HTHH effect.

**Response:** We acknowledge the reviewer's concern about the apparent positive bias in MIAWARA-C measurements relative to MLS during 2019-2020. This deviation from the typical negative bias warrants careful consideration given its magnitude.

The only instrumental problem that fits with this timeline, was that a window on the instrument had broken, potentially allowing more moisture than normal to enter the instrument, and, though this could be responsible for larger noise in the measurements due to potentially greater attenuation of the middle atmospheric signal, we see no reason for this to cause the MIAWARA-C minus Aura-MLS bias to switch from being negative to positive.

In contrast to the theory that this is an instrumental problem, one author of this paper is currently investigating an anomaly in wind observations from several locations across northen europe. The

period coincides with unusual dynamical conditions in the Arctic middle atmosphere. Specifically, we observe disrupted seasonal wind transitions in co-located meteor radar observations across northern Europe, suggesting anomalous transport patterns that could affect water vapor distributions. The 2019-2020 winter featured an exceptionally strong polar vortex followed by dynamic disturbances, which may have created localized water vapour enhancements not fully captured by MLS's sampling at this location.

While we cannot definitively attribute the 2019-2020 anomaly to either instrumental or geophysical causes, we note that the anomaly predated the Hunga eruption by 2-3 years, and after 2021 measurements again agreed well with MLS. We would also like to not that the climatological change in heating rates changed the concluded heating rate anomalies by less than 15% in a sensitivity study, when the years 2019 and 2020 were excluded from the climatology.

[Figure]

*Meridional and zonal winds in the MLT region from Esrange, Sweden, and Alta, Norway.*

**Added:** We note that MIAWARA-C measurements show an anomalous positive bias relative to MLS during 2019-2020, possibly related to unusual Arctic dynamics during this period. While the cause remains under investigation, sensitivity tests show this does not significantly affect our post-Hunga heating rate calculations, which rely primarily on 2023-2024 data when agreement with MLS is improved.

**Comment:** What are the co-location criteria for MLS?

**Response:** The nearest limb-tangent points for each half-day are used. This is less than 1km for both Zimmerwald and Ny-Ålesund.

**Added** [line 163]:For the MLS data, the nearest limb-tangent point to each radiometer location is used in this study.

**Comment:** Line 154: "The observations from Zimmerwald, BE, show that already in summer of 2022, water vapour mixing ratios at 0.1 hPa are above preceding years".

Is 0.1 hPa correct, or is it 1 hPa as in Figure 1?

**Response:** Correct

**Changed** [line 297]**:**The observations from Zimmerwald, BE, show that already in summer of 2022, water vapour mixing ratios at 1 hPa.

**Comment:**159 – 164: It would of course be very good to know what this bias is due to? And how it can be corrected for. The bias is larger than the HTHH H2O anomaly in the last year, so this is a major problem in my opinion. How can we trust the measurements in 2023 and 2024?

**Response:** We acknowledge the reviewer's concern about the apparent positive bias in MIAWARA-C measurements relative to MLS during 2019-2020. This deviation from the typical negative bias warrants careful consideration given its magnitude.

The only instrumental problem that fits with this timeline, was that a window on the instrument had broken, potentially allowing more moisture than normal to enter the instrument, and, though this could be responsible for larger noise in the measurements due to potentially greater attenuation of the middle atmospheric signal, we see no reason for this to cause the MIAWARA-C minus Aura-MLS bias to switch from being negative to positive.

In contrast to the theory that this is an instrumental problem, one author of this paper is currently investigating an anomaly in wind observations from several locations across northen europe. The period coincides with unusual dynamical conditions in the Arctic middle atmosphere. Specifically, we observe disrupted seasonal wind transitions in co-located meteor radar observations across northern Europe, suggesting anomalous transport patterns that could affect water vapor distributions. The 2019-2020 winter featured an exceptionally strong polar vortex followed by dynamic disturbances, which may have created localized water vapour enhancements not fully captured by MLS's sampling at this location.

While we cannot definitively attribute the 2019-2020 anomaly to either instrumental or geophysical causes, we note that the anomaly predated the Hunga eruption by 2-3 years, and after 2021 measurements again agreed well with MLS. We would also like to not that the climatological change in heating rates changed the concluded heating rate anomalies by less than 15% in a sensitivity study, when the years 2019 and 2020 were excluded from the climatology.

[Figure]

*Meridional and zonal winds in the MLT region from Esrange, Sweden, and Alta, Norway.*

**Added** [line 308]**:** We note that MIAWARA-C measurements show an anomalous positive bias relative to MLS during 2019-2020, possibly related to unusual Arctic dynamics during this period. While the cause remains under investigation, sensitivity tests show this does not significantly affect our post-Hunga heating rate calculations, which rely primarily on 2023-2024 data when agreement with MLS is improved.

**Comment:**
Figure 2, right panels: There seem to be jumps and biases in the MIAWARA-C H2O mixing ratios at 1 hPA, e.g., very large values in early 2024, and a discontinuity of about 2 ppm from end of December 2023 and beginning of 2024, a 1 ppm decrease around day 300, and the large values in 2019 or 2020. I think these signatures should be understood and corrected for. Also, no error bars are given and errors are not discussed at all.

**Response**: Regarding the discontinuities in the MIAWARA-C time series - this was investigated in the original data analysis of the water vapour time series for the MIAWARA-C. The jump at the beginning of 2024 appears to be larger than the rest of the time series for several reasons. The first is of the way in which the data is presented. The presented lines in fact do not show the daily data, but a 3 day moving average (this is now stated explicitly in the manuscript). The way this was employed, however, did not require a minimum number of observations in the 3 day moving window. In the case of the start of 2024, there was no converged retrieval for the first day of 2024. This meant that the first day of 2024 is treated as a daily observation and appears as a spike, due to the also very high water vapour mixing ratios in this period. When the two dimensional (time vs pressure) data is seen, the data appears vastly less discontinuous, see the figure on page 3.

Regarding the apparent larger variance and increased bias in 2019 : please see earlier comments and discussion on the measurement with reference to wind measurements.

Regarding the error analysis, this is a good point and something that has been added to the paper.

**Change**: Figure 2 has now been changed so that two points are required for a moving average. It has been added to the figure description explicitly that this is a three day moving average.

See section 3.4 for discussion on retrieval error uncertainty propogation into the heating rates and flux estimates.

Comment: Line 174: "can remain there for much longer, on the order of years to decades,"

Multiple decades are probably unrealistic, right?

**Change:** rephrased [line 421] : it is generally agreed that higher than average total middle atmosphere water vapour will persist for between five and ten years following the eruption (Schoeberl et al., 2022; Jenkins et al., 2023; Stocker et al., 2024; Millan et al., 2022).

**Comment:**
Line 178: "(by approximately SI25 %)"

Please explain what SI 25% means.

**Response:** this is a formatting error

**Change: correction:** (by approximately 25%)

**Comment:** Radiative forcing is typically determined at the tropopause level.

**Response:** I would not say this is the always the case. For example, the IPCC convention is to do so for the top of atmosphere, and for some applications the surface is more pertinant (IPCC, 2013, Chapter 8, p. 659–740).

Line 187: "It is common in flux calculations for climate studies to calculate changes to downward fluxes at the surface (with the obvious application of surface heating rates)"

**Comment:** Line 182: "and at the lower measurement response limit of the microwave radiometer observations at 10 hPa."
This should have been mentioned before. What is the lowest pressure level measurable?

**Response:** With radiometric measurements, quite often it is the case that sensitivity to different pressure levels gradually drops off, so a decision about what the minimum measurement response you want to take into consideration is- for some studies this is 60% (e.g. Stähli et al. (2013)). Where the averaging kernels are discussed, the pressure level range where the measurement response is above 0.6 is also commented. 10 hPa was found to be a suitable reference height to make calculations at for the two radiometers.

**Change: Added in MIAWARA section:** The mean range for which the retrievals have measurement response of over 60% is from 0.034 hPa to 3.6 hPa.

**Added in MIAWARA-C Section: ...** and the mean measurement response above 60% between 7.5 hPa and 0.02 hPa.

**Comment:** Line 187: "the longwave fluxes at the top of the troposphere could be more pivotal in determining global climate patterns than those at the surface." "could" or is it actually the case? The statement is quite weak and I'm not sure what its intention is?

**Response:** Agreed, this sentence didnt seem to add mch so it has been removed.

**Comment:** Line 222: "The temperature parameter decreases with each iteration,"

What is the "temperature parameter"?

**Response:** This is described in greater detail in the referenced paper, which has been made more obvious for the reader

**Modified** [line 222]**:** The temperature parameter (not to be confused with any physical or brightness temperature) controls the likelihood of accepting sub-optimal solutions: at high temperature, even considerably worse solutions have a reasonable chance of being accepted, encouraging wide exploration of the solution space (see more in Kirkpatrick et al., (1983)).

**Comment:** Line 229. "The total flux is then found by integrating over all frequencies and 15 elevation angles between 0 (zenith) and 90 (horizon)."

I guess you also have to integrate azimuthally, i.e., the integration is actually done over solid angles, not only the elevation angle.

**Response:** The azimuthal component is included in the flux integration; however, due to the assumed azimuthal symmetry of the atmosphere in our simulations, radiative transfer calculations are only performed for a single azimuth. The resulting radiances are then weighted accordingly over the full $2\pi$ azimuthal range when integrating over the solid angle.

**Comment:** Line 249: "and the full ARTS method"

The "full ARTS method" is method 1 above, right? It would be good to introduce the term "full ARTS method" in the section above.

**Modified:** Changed to 'simulated annealing method' here and elsewhere.

**Comment:** Line 255: "standard deviation of" -> "standard deviation is"?

**Response:** As we talk about a dataset, I think it makes more sense to talk about the standard deviation **of** said dataset.

**Comment:** Line 256: "Despite this, the anomalies predicted by both methods, by comparing fluxes simulated from the climatology to fluxes calculated in the post-eruption period, a very good

agreement is found. Throughout the period, the entire ARTS method showed more intense fluxes by a mean of 0.0017Wm-2,"

How is this possible? If the flux calculations for the two approaches differ by up to 0.5 W/m2, how can the differences wrt to the reference period be so small. This cannot be true. Or the description here is wrong or misleading? Or I am missing a point ...

**Response:** The figure is perhaps a little misleading. The anomalies were calculated with respect to each method, leading to smaller anomaly differences than the difference in absolute flux. The height used for this method comparison was also well below the 60% measurement response limit of the radiometer, at 14km. This means that the influence of the a priori profile on the fluxes is very significant, thus a large component of the $-18Wm^{-2}$ comes from that a priori profile, which in this comparison is only climatological and thus resultls in unchanged anomalies. When the anomaly differences are compared to the anomaly values, the difference is aound 10%.

**change**: added clause [line 257]: Despite this, the anomalies predicted by both methods, by comparing fluxes simulated from the climatology to fluxes calculated in the post-eruption period (with each method calculating the climatology and the post-eruption period), a very good agreement is found.

**Comment:** Line 271: "The propagation of solar fluxes is …and thus IS .."

**change:** As suggested

**Comment:** Line 272: "but also highly seasonal dependent at high (arctic) and low antarctic latitudes."

**change:** Added [line 322]:  The reduction in shortwave flux has a seasonal dependence due to the difference in the top of atmosphere down-welling solar flux throughout the year, as well as the observations, presented having the general characteristic of a greater water vapour mixing ratio anomaly in summer compared to winter.

**Comment:** Line 273: "A computation .. was also calculated"

Sounds a bit odd.

**change:** A computation of change in the mean down-welling short wave fluxes was also **made**

**Comment:** Line 276: "CO_4"??

Does this exist? Or do you mean methane?

**change:** As suggested

**Comment:** Line 277: "Coddington et al. (2017)"

Wrong cite command.

**change:** As suggested

**Comment:**Line 296: ".. the retrievals from Ny Alesund TAKE"

**change:** As suggested

**Comment:** Same sentence: "take a lower a priori profile"

"Lower profile" can have different meanings.

**change:** rephrased paragraph to improve clarity [line 313]: A key difference in the retrieval techniques between the measurements from Zimmerwald and Ny-Ålesund lies in the treatment of the a priori water vapour profile. For Zimmerwald, a static a priori profile is used, derived from a climatological dataset combining ECMWF IFS analyses with MLS observations. In contrast, the Ny-Ålesund retrievals use a time-dependent a priori: the lower part of the profile is taken from the ECMWF analysis closest to the MIAWARA-C observation time, with a blended transition between ECMWF and MLS data in the range 0.02-0.5 hPa, and ECMWF data only below 0.5 hPa. This means that below the lower measurement response limit of the MIAWARA-C (typically below 30 km or 10 hPa), water vapour profiles may deviate from the climatology even if there is a significant error in the a priori profile. For this reason, downwelling fluxes were calculated again at 10 hPa at Zimmerwald and  Ny-Ålesund.

**Comment:** Line 301: "The anomalies presented in figure 5 show a positive response (meaning less downwelling flux) for"

The flux has not been properly defined above and this should be done (see my earlier comment). As far as I know the standard definition is downward – upward, implying that a positive net flux leads to more downwelling radiation.

**Response**: This is now defined in section 2.4. For the upward fluxes at the lower threshold (10 hPa), we assume unchanged conditions as our observations do not cover this part of the atmosphere, so they remain unchanged. Similarly at the toa, downwelling fluxes are unchanged (zero for longwave, defined by date and latitude for shortwave). The convention used is perhaps more common in radiaitve transfer theory than climate studies (Buglia, 1986).

**Change**: [line 198] In this study, down-welling fluxes are assigned a negative sign  and up-welling fluxes a positive sign . Accordingly, an increase in downward flux appears as a more negative value, whereas an increase in upward flux appears as a more positive value.

**Comment:** Line 312: "In contrast to the downward flux measurements"

Why "Downward flux measurements" ? You did not use flux measurements, did you.

**Change**: correction: downward flux simulations

**Comment:** Figure 5: These are not the fluxes, but the flux anomalies, right? Both the caption and the y-axis label state that fluxes are shown. This is not correct, as far as I can tell?

**Change**:  The short-wave and long-wave up-welling upwelling flux anomalies

**Comment:** Line 319: "By contrast, at Ny-Ålesund, the bulk of the water vapour increase extends to lower altitudes (and cooler stratospheric conditions), thereby inhibiting the net emission of energy to space."

I am wondering at what altitude the H2O absorption bands really become optically thin?

**Response**: Top of atmosphere to an optical depth of 0.01 is reached between 1 hPa and 10 hPa

Figure 6, top panel: The shortwave flux is (basically) zero, but the curves for the LW and the sum are not identical. Something must be wrong here.

**Change**:  plots have been corrected

Figure 6, top panel: I'm not sure I understand fully, why the SW flux anomaly is zero at the TOA, but not in the lower atmosphere. Perhaps you can explain this briefly.

**Response**:
Under clear-sky conditions, the short-wave anomaly at the TOA is near zero because the only affected term there- the upwelling reflected component- is small due to low surface albedo and strong atmospheric attenuation along the upward path, whereas the stratospheric downwelling beam remains large and though it is small, there is a visible anomaly caused by the water vapour.

Both panels of the Figure: again, the figures show flux anomalies, not fluxes themselves?

**Change**: correction: The short-wave and long-wave down-welling flux anomalies

**Comment:** Line 327: "The mesosphere exhibits an atmospheric circulation from the summer pole to the winter pole, as air rises from the summer pole,"

The meridional circulation from summer to winter hemisphere is not a consequence of the upwelling above the summer pole and the downwelling at the winter pole, but caused by breaking gravity waves.

**Response**: Yes, this wasn't meant to demonstrate causality but that doesn't seem clear in this sentence.

**Change**: correction [line 347]: The mesosphere exhibits an atmospheric circulation from the summer to the winter hemisphere driven primarily by momentum deposition from breaking gravity waves, with additional modulation by tides and planetary waves. This forcing induces up-welling over the summer pole and down-welling over the winter pole.(Lindzen, 1981; Smith, 2012; Becker, 2012).

Figure 7, bottom panel: How do you deal with the high bias of the Ny-Ålesund measurements during the two years before the eruption? They will directly affect the heating rate anomalies, i.e., the heating rate anomalies for Ny-Ålesund will be systematically wrong and the results shown are not reliable. Or am I missing a point?

**Response**:

Please see earlier comments with reference to the high bias with respect to Aura-MLS measurements. In the study, we trust the MIAWARA-C measurements from this time period, and so have included them when calculating the heating rate and flux anomalies.

**Comment:** Figure 8, bottom panel: From this depiction it is not possible to tell what part of the signatures is related to HTHH H2O. What about interannual variability and the high bias in H2O at Ny-Ålesund in 2019 and 2020?

**Response**: This is a good point. The large "bias" in 2019 was included in the simulations, so if it were to transpire that there was in fact a problem with the measurements in this 18 month period where measured values are above Aura-MLS, the anaomaly would be larger.

comment: Figures 7 and 8: The pressure range sensitive to H2O is of course important and should be mentioned.

**Change:** added in the caption this information: measurement sensitivity range added to captions: The heating rate anomaly after the Hunga eruption as a result of the increased short-wave absorption above Zimmerwald (top) [measurement sensitivity between 3.6 hPa and 0.034 hPa] and Ny-Ålesund, Svalbard (bottom) [measurement sensitivity between 7 hPa and 0.02 hPa] between 2022 and 2024.

**Comment:** Line 342: "Shortwave absorption by water vapour molecules is substantially .."

So, the above and Fig. 7 only considers the LW effect. This must of course be mentioned in the paragraph above.

**Response**: Actually, the figure 6 and 7 include shortwave conttributions (previously mentined orange line). But the shortwave emission by atmospheric water vapour, which can be considered to be negligable at mesospheric/stratospheric physical temperatures, was not included. Only absorption of incoming solar radiation is considered.

**Change** [line 362]**:** This paragraph has been modified to improve clarity:Shortwave absorption by water vapour molecules is substantially weaker than their long-wave absorption. Nevertheless, to fully assess the total radiative heating-rate anomaly (as has been done in the analysis of flux anomalies), we include the short-wave contribution alongside the long-wave heating rates. Because water vapour emission in the short-wave can be considered negligble for all purposes in this study, at middle-atmospheric temperatures, short-wave emission was neglected in the radiative-transfer model- only absorption of incoming solar irradiance was considered.

**Comment:** Line 351: ".. the combined effect .. is combined"

**Change:** removed second 'is combined'

Reviewer 2

**Comment:** This study assesses the water vapour anomalies observed over Zimmerwald (Switzerland) and Ny Alesund (Norway), due to the Hunga eruption in 2022, and their subsequent radiative effects (radiative flux changes, heating rates). I associate with Anonymous Reviewer #1's general remarks and her/his 4 Major Comments, that I recommend to address. I add a general comment on the structure of the paper. I think that the manuscript, in the way it is structured at present, is very confusing for the reader due to mixing up methods, data, results, interpretations. I strongly suggest to structure the manuscript with a classical suite: Introduction, Data and Methods, Results and Discussion, Conclusions.

**Response:** Thanks for this feedback. We agree that the structure of the paper was not very clear and this has now been altered as suggested.

Also, I found it very hard to find the actual quantitative results (which are additionally not even mentioned in the Abstract), which I had to go search for in the Conclusions. Please try to be more factual and synthetic in the Results section, so to put your (interesting) results more in value. Plase also find a few specific comments in the following. Despite these points, the results are interesting and worth being published in ANGEO, once these points are resolved. Quantitative estimations of the water-vapour-related heating (cooling, in fact) rates are particularly important due to the lack of literature on this. Thus, another major comment for me would be to highlight the importance of heating rates estimations for such stratospheric events, in the Introduction (this is not the case in the present version of the manuscript).

**Response:** Yes, we agree that the results were not presented in a manner which was easy to find concrete numbers outside of the plots presented in the results section and inside the conclusion section.

**Change:**

-Added quantitative statements to the abstract in line with this comment [line 426]: to the Between March 2023 and November 2024, long-wave (LW) down-welling fluxes at 10 hPa increased by 0.010 W m−2 to 0.040 W m−2 on > 90% of days above Zimmerwald...Net LW + SW heating-rate anomalies averaged over 1 January 2023 - 1 January 2025 were −0.012 K d−1 at 1 hPa10 above Zimmerwald and −0.016 K d−1 at 1 hPa above Ny-Ålesund. Although the additional radiative forcing at the surface is modest ($\sim$ 0.02 W m−2, or $\sim$ 1 % of the anthropogenic CO2 forcing accumulated since 1750),

-Added section titled "Radiative fluxes and local heating rate implications" after general introduction [line 50]

-Added section: 3.4 Uncertainty assessment to give quantitative estimations of the uncertainty about the simulated heating rate/flux estimations [line 380]

**Comment:** 1) Title: consider the possibility to mention which are the two locations

-Changed to: Radiative impact of increased middle atmospheric water vapour in the aftermath of the Hunga 2022 volcanic eruption at Zimmerwald, Switzerland and Ny-Ålesund, Svalbard

**Comment:** Abstract: please add some quantitative results in the abstract (e.g. what is mentioned in the Conclusions), for the moment there aren't any

**Response**: Agreed, please see above comment regarding adding more quantitative data

**Comment:** L11: "would have been" --> "is"

-Change: As suggested

**Comment:** L14: is "launching" the good word here?

-Changed: to "triggering"

**Comment:** L10: add a reference for this estimation

-Added: (Millan et al., 2022)

**Comment:** L22: with "stable" you mean that has a long lifetime?

**Response**: Yes, maybe saying it is stable leads to the implication that it is chemically stable which is not correct.

-Changed: to "is relatively long-lived"

L27-28: This injected amount has been revisited to slightly larger values recently (1.0 Tg as a lower bound), please refer to: https://agupubs.onlinelibrary.wiley.com/doi/full/10.1029/2023GL105565

**Response**: Thanks for this information

-Changed: to " a moderate amount (totalling 1.6 ± 0.5 Tg) of sulphur dioxide was also emitted during the entire eruption sequence (Sellitto et al., 2024)."

**Comment:** 8) L28-31: add references

-Added references here: Whilst increased water vapour has a warming net radiative effect at the surface due to increased long-wave absorption, sulphur dioxide, which forms sulfate aerosols in the atmosphere, has the effect of reducing the amount of short-wave radiation that reaches the surface and, thus, results in a negative radiative impact at the surface (Quaglia et al., 2023; Wang and Dickinson, 2013).

**Comment:** L32-33: "meaning that...limit": this is not completely true (as we're already beyond 1.5°C limit, unfortunately) but Jenkins et al. just affirm that the probability to continuing exceeding this threshold might be slightly increased by water vapour from Hunga. Please correct this statement.

-Change:removed the  "meaning that...limit" statement

**Comment:** L32-33:

10) L34-35: yes, but please also note that Sellitto et al. found a positive top-of-atmosphere radiative forcing at the very beginning of the plume dispersion - which is worth mentioning

-Added to sentence [line 59] : despite a slightly positive forcing at the top of atmosphere, with the total effect of water vapour plus aerosol resulting in a 0.2 W m$^{-2}$ increase in the radiative forcing (downward minus upwards).

If heating rates are also studied in this work, why the existing heating rates estimations for Hunga (water vapour and aerosols, see e.g. Sellitto et al., 2022) are not discussed in the introduction? Also, as heating rates of stratospheric perturbations are a not-yet-well-studied topics (see e.g. https://acp.copernicus.org/articles/23/15523/2023/) why this is not discussed more in the Introduction? Also, please stress the fact that here you are just dealing with water vapour impacts and the impact from Hunga's aerosols is not considered

**Response**: Good point. The introduction has been split into two sections now, one includes a description of heating rates.

Added (section 1.2 line 65): These vertically resolved stratospheric heating rates, however, remain sparsely quantified compared with surface/TOA radiative forcing and even recent event studies have only begun to estimate them directly (e.g. first hemispheric and in-vortex HRs for the 2019–2020 Australian smoke plume) (Sellitto et al., 2022a)

**Comment:** L90: The ARTS RTM is introduced here but I wonder if a specific section, with more details on this RTM, would be needed in a Data and Methods section

**Added** [line 176]: ARTS, the radiative simulation software used in this study, is a line-by-line radiative transfer model that provides accurate calculations of atmospheric absorption and emission by computing radiative properties across individual spectral lines rather than using parameterised approximations (Eriksson et al., 2011). This line-by-line treatment offers higher spectral fidelity for diagnosing composition effects on radiative forcing than the parameterised schemes used in climate models, albeit at higher computational cost (Buehler et al., 2005). The latest release (v2.6) features deep Python integration, supports fully polarised radiative transfer in spherical 1D-3D geometries, and includes a built-in optimal-estimation retrieval module (Buehler et al., 2025). In addition, ARTS now covers the solar/shortwave spectrum (including the visible), with newly implemented shortwave solvers and evaluations reported by Brath et al. (2024) and described in the v2.6 overview (Buehler et al., 2025).

Title of Sect. 2.2: "Space-borne" adjective lacks a noun ("observations"? So, it would rather be "Space-based observations")

**Change:** As suggested

Section 3: I'm missing here an explicit title "Results" (with subsections)

**Change:** section title named "Results and Discussion" added

L132: Among the references for water vapour plume evolution, clearly this one is missing: https://www.nature.com/articles/s43247-022-00652-x

**Change:** Added as suggested

L134-135: Decisive for the dispersion path of the plume as a whole, and happening at the very beginning of the plume dispersion, before what's mentioned here, is the radiatively driven descent described by Sellitto et al., 2022, which is associated with the radiative cooling of the plume for infrared emission from water vapour. Please add this here.

**Change:** All plume evolution discussion is now in the introduction. In section, 1.1 the following sentence has been added to the start of paragraph 3:
 At the very beginning of the dispersion, strong infrared emission from the vapour-rich core caused net radiative cooling that drove a rapid, radiatively driven descent of the plume by several kilometres, decisively shaping its subsequent dispersion pathway (Sellitto et al., 2022b).

172: "such as the one..." this seems to suggest that Hunga injected in the troposphere, please clarify

**Response**: In the eruption, water vapour from the plume was also deposited in the troposphere as as well as the stratosphere, just that this is not remarkable like it was for the stratospheric injection.

18) L175: "years to decades" where these are taken from (add a reference, please)? Also, "decades" sounds like a bit too much.

**Response:** This was taken from the study referenced by Jucker et al., but you are correct- projections do not span to multiple decades.

**Change:**  This means that, when a rare event occurs, such as the injection of water vapour into the middle atmosphere (as was the case with the Hunga eruption) perturbations can have effects lasting for several years. Jucker et al. (2023) have found from climate model simulations that the increased water vapour from the Hunga volcano is likely to last eight years or more.

L178: what does "SI25%" mean?

**Response:** Formatting error with siunix package

**Change**: 25%

Section 4.2: It sounds odd to me to have a Methods section here. I suggest to aggregate this to the "Instrument" section, i.e. in a comprehensive "Data and Methods" section. Like it is, this confuses a lot the reader.

**Response:** Agreed, this has been changed now and several paragraphs have been moved to accomodate the new structure

Change:  Section 2 is now titled: Data and Methods

21) L360: the beginning of the Conclusion sounds strange to me, because nowhere the Hunga event is mentioned (e.g. "the anomaly of water vapour..." due to what?). It is clear that you are talking about the Hunga anomalies but please produce self-consistent Conclusions

**Change** [line 420]: slight modification to:  Whilst uncertainty persists about how long the anomaly of water vapour in the middle atmosphere will persist following the January 2022 Hunga eruption, and the global distribution of this increase in water vapour, it is generally agreed that higher than

average total middle atmosphere water vapour will persist for between five and ten years following the eruption (Schoeberl et al., 2022; Jenkins et al., 2023; Stocker et al., 2024; Millan et al., 2022)

22) L362: "2027", "2032", this is very quantitative. Are there references that actually mention these specific years? and demonstrate this? I would be less quantitative here, due to the mentioned high degree of uncertainty

**Response:** This was based off of projections of "5-10 years", but indeed it does sound more precise than it is possible to be accurate. The sources between them give several estimates which range from the 5 to 10 year projections.

**Change** [line 420]: slight modification to:  Whilst uncertainty persists about how long the anomaly of water vapour in the middle atmosphere will persist following the January 2022 Hunga eruption, and the global distribution of this increase in water vapour, it is generally agreed that higher than average total middle atmosphere water vapour will persist for between five and ten years following the eruption (Schoeberl et al., 2022; Jenkins et al., 2023; Stocker et al., 2024; Millan et al., 2022)

References

Bell, A., Sauvageat, E., Stober, G., Hocke, K. and Murk, A., 2025. Developments on a 22 GHz microwave radiometer and reprocessing of 13-year time series for water vapour studies. Atmospheric Measurement Techniques, 18, pp.555-567.

Buglia, J.J., 1986. Introduction to the theory of atmospheric radiative transfer. NASA Technical Memorandum, NAS 1.61:1156.

Forkman, P., Eriksson, P. and Winnberg, A., 2003. The 22 GHz radio-aeronomy receiver at Onsala space observatory. Journal of Quantitative Spectroscopy and Radiative Transfer, 77, pp.23-42.

IPCC, 2013. Climate Change 2013: The Physical Science Basis. Contribution of Working Group I to the Fifth Assessment Report of the Intergovernmental Panel on Climate Change. Chapter 8: Anthropogenic and Natural Radiative Forcing, pp.659-740. Cambridge University Press, Cambridge, United Kingdom and New York, NY, USA.

Kirkpatrick, S., Gelatt Jr, C.D. and Vecchi, M.P., 1983. Optimization by simulated annealing. Science, 220, pp.671-680.

Niemeier, U., Rieger, L., Timmreck, C., Clyne, M., Hoffmann, L. and Bourassa, A., 2023. Volcanic stratospheric injections up to 160 Tg(S) yield a Eurasian winter warming indistinguishable from internal variability. Journal of Geophysical Research: Atmospheres, 128, e2023GL106482.

Quaglia, I., Timmreck, C., Niemeier, U., Visioni, D., Pitari, G., Brodowsky, C., Brühl, C., Dhomse, S.S., Franke, H., Laakso, A., et al., 2023. Interactive stratospheric aerosol models' response to

different amounts and altitudes of SO2 injection during the 1991 Pinatubo eruption. Atmospheric Chemistry and Physics, 23, pp.921-948.

Sellitto, P., Belhadji, R., Kloss, C. and Legras, B., 2022a. Radiative impacts of the Australian bushfires 2019–2020–Part 1: Large-scale radiative forcing. Atmospheric Chemistry and Physics, 22, pp.9299-9311.

Sellitto, P., Podglajen, A., Belhadji, R., Boichu, M., Carboni, E., Cuesta, J., Duchamp, C., Kloss, C., Siddans, R., Bègue, N., et al., 2022b. The unexpected radiative impact of the Hunga Tonga eruption of 15th January 2022. Communications Earth & Environment, 3, 288.

Sellitto, P., Siddans, R., Belhadji, R., Carboni, E., Legras, B., Podglajen, A., Duchamp, C. and Kerridge, B., 2024. Observing the SO2 and sulfate aerosol plumes from the 2022 Hunga eruption with the Infrared Atmospheric Sounding Interferometer (IASI). Geophysical Research Letters, 51, e2023GL105565.

Stähli, O., Murk, A., Kämpfer, N., Mätzler, C. and Eriksson, P., 2013. Microwave radiometer to retrieve temperature profiles from the surface to the stratopause. Atmospheric Measurement Techniques, 6, pp.2477-2494.

Wang, K. and Dickinson, R.E., 2013. Global atmospheric downward longwave radiation at the surface from ground-based observations, satellite retrievals, and reanalyses. Reviews of Geophysics, 51, pp.150-185.